# Towards Robust and Efficient Cloud-Edge Elastic Model Adaptation via Selective Entropy Distillation

**Yaofo Chen**[12][*]**Shuaicheng Niu**[3*]**, Yaowei Wang**[2*]**, Shoukai Xu**[1]**, Hengjie Song**[1]**, Mingkui Tan**[145†]
South China University of Technology[1] Pengcheng Laboratory[2] Nanyang Technological University[3]
Key Laboratory of Big Data and Intelligent Robot, Ministry of Education[4] Pazhou Laboratory[5]
`chenyaofo@gmail.com; mingkuitan@scut.edu.cn`

## Abstract

The conventional deep learning paradigm often involves training a deep model on a server and then deploying the model or its distilled ones to resource-limited edge devices. Usually, the models shall remain fixed once deployed (at least for some period) due to the potential high cost of model adaptation for both the server and edge sides. However, in many real-world scenarios, the test environments may change dynamically (known as distribution shifts), which often results in degraded performance. Thus, one has to adapt the edge models promptly to attain promising performance. Moreover, with the increasing data collected at the edge, this paradigm also fails to further adapt the cloud model for better performance. To address these, we encounter two primary challenges: 1) the edge model has limited computation power and may only support forward propagation; 2) the data transmission budget between cloud and edge devices is limited in latency-sensitive scenarios. In this paper, we establish a Cloud-Edge Elastic Model Adaptation (CEMA) paradigm in which the edge models only need to perform forward propagation and the edge models can be adapted online. In our CEMA, to reduce the communication burden, we devise two criteria to exclude unnecessary samples from uploading to the cloud, *i.e.*, dynamic unreliable and low-informative sample exclusion. Based on the uploaded samples, we update and distribute the affine parameters of normalization layers by distilling from the stronger foundation model to the edge model with a sample replay strategy. Extensive experimental results on ImageNet-C and ImageNet-R verify the effectiveness of our CEMA.

## 1 Introduction

Deep neural networks (DNNs) have witnessed remarkable breakthroughs in a broad spectrum of applications from computer vision (He et al., 2016; Dosovitskiy et al., 2021) to natural language processing (Radford et al., 2018; Brown et al., 2020). In real-world applications, the traditional deployment pipeline of DNNs is as follows: 1) training a large/foundation model on a cloud server and 2) distilling/compressing the large/foundation model into a smaller model to be deployed in edge devices for delay-sensitive applications. This pipeline has gained great success when test samples share the same distribution as the training ones. However, in real-world edge devices, the environment may dynamically change and the distributions of test samples are different from training ones. Such distribution shifts often result from natural variations or corruptions, such as changes in lighting and sensor degradation (Hendrycks & Dietterich, 2019; Koh et al., 2021). In this case, models may exhibit significant performance drop (Wang et al., 2021; Zhang et al., 2022a).

To handle the distribution shift, previous methods seek to update the edge model, which can be roughly categorized into two groups: i) *Offline generalization* methods are executed on the cloud and then distribute updated models to the edge devices. Specifically, unsupervised domain adaptation methods (Zhang et al., 2020; Liang et al., 2020; Qiu et al., 2021; Lin et al., 2022) perform model adaptation on collected test data in an offline manner. Domain generalization methods (Li

---

[*]Equal contribution. [†]Corresponding author.

Figure 1: Comparisons between the conventional Test-time Adaptation (TTA) (left) and our Cloud-Edge Elastic Model Adaptation (right). The conventional one locally performs adaptation only in the edge with limited resources. In contrast, our CEMA conducts model adaptation more efficiently in the edge, which offloads the heavy adaptation workloads to the cloud with massive resources.

et al., 2018; Dou et al., 2019) pre-anticipate the possible test shifts at the training time, in which the possible shifts can be simulated by a meta-learning scheme. However, they may yield inferior performance since it is hard to pre-anticipate all unknown shifts at training time. ii) *Online generalization* methods directly learn the shifts by adapting the model with test data. Recently, test-time training (Sun et al., 2020; Bartler et al., 2022) and fully test-time adaptation (TTA) methods (Wang et al., 2021; Niu et al., 2022; 2023) are newly devised to adapt a model to the test domain in an online manner, which are more practical in real-world applications. However, they may be computationally heavy to perform back-propagation, which may be unaffordable in resource-limited edge devices.

Besides, the foundation model in the cloud also should be continuously updated using the test samples in the edges. To address the above issues, one can leverage both the cloud and the edge by uploading all the test samples to the cloud for adaptation of both the foundation and edge models. However, it is still very challenging: 1) The data communication burden between the cloud and edges may be heavy. Since the communication overhead is mainly affected by the number of uploaded samples. It not only decreases the adaptation efficiency in the cloud but also consumes the limited bandwidth in the cloud-edge system. 2) How to exploit the foundation model to enhance the performance of the edge model on distribution-shift test data is an open question. Typically, the cloud has much richer computational resources and budgets than edges. In this case, the cloud is able to support heavier computation and leverage more complex and stronger models for adaptation.

In this paper, we propose a Cloud-Edge Elastic Model Adaptation (CEMA) paradigm that executes dynamic model adaptation in a cloud-edge collaborative way instead of inference with the fixed model. As shown in Figure 1, we delegate all adaptation workloads to the cloud and thus only require vanilla inference in edges. To reduce communication overhead, we exclude two types of samples from uploading to the cloud: 1) unreliable samples with high entropy identified by a dynamic entropy thresholding scheme; 2) low-informative samples with low entropy identified by an unchanged thresholding scheme. Based on this, our CEMA greatly reduces the communication burden. To leverage rich knowledge in the foundation model, we use it to guide the edge model via knowledge distillation for adaptation. To improve the data utilization efficiency of uploaded samples, we devise a replay buffer to store and reuse these samples. We distill the foundation model to the edge model based on both the newly uploaded samples and samples from the replay buffer. In this way, our CEMA achieves better performance than the vanilla adaptation.

**Main novelty and contributions**: 1) We establish a Cloud-Edge Elastic Model Adaptation (CEMA) paradigm designed for efficient collaborative model adaptation. Our CEMA is a general paradigm that is applicable to online adapt edge models to new dynamically changing environments. 2) We improve the adaptation performance of the edge model by performing a replay-based entropy distillation, which minimizes prediction entropy and the KL divergence between the edge model and the foundation model using a sample replay strategy. 3) We reduce communication costs by devising entropy-based criteria for excluding unreliable and low-informative samples from being uploaded. Experimental results show CEMA lowers 60% communication cost than SOTAs on ImageNet-C.

## 2 CLOUD-EDGE COMMUNICATION-EFFICIENT MODEL ADAPTATION

**Problem statement**. In this paper, we focus on how to efficiently improve adaptation performance in the context of cloud-edge deployment under distribution-shifted scenarios. Let $g_w(\cdot)$ denote a model trained in a powerful cloud server on a set of training data. Instead of deploying the model $g_w(\cdot)$ in the cloud, we would infer $g_w(\cdot)$ locally on a resource-limited edge device (*e.g.*, a surveil-

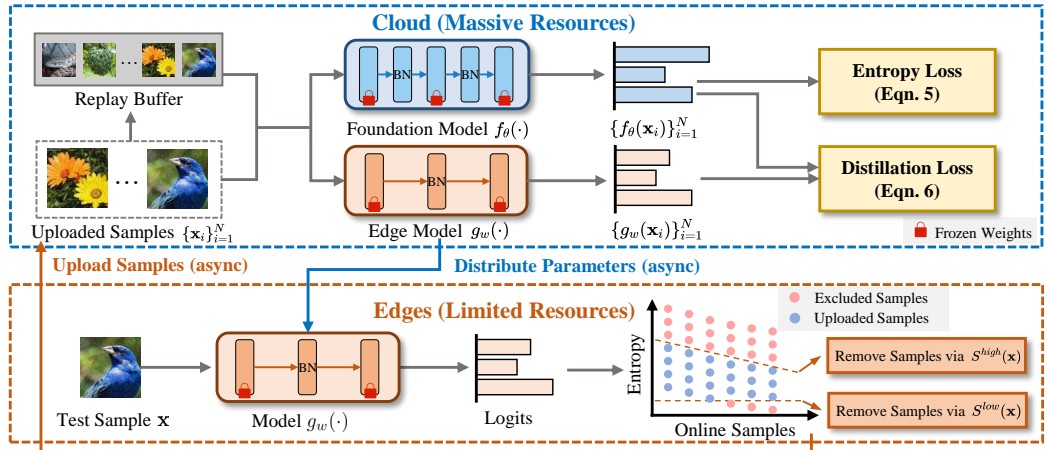

Figure 2: An overview of our proposed CEMA. In edge: after inference, each edge asynchronously uploads samples to the cloud by excluding unreliable ones (based on $S^{high}(\mathbf{x})$ in Eqn. (1)) and low-informative ones (based on $S^{low}(\mathbf{x})$ in Eqn. (3)). In cloud: 1) our CEMA improves the foundation model $f_\theta(\cdot)$ with the uploaded samples via entropy minimization (Eqn. 5) and meanwhile stores uploaded samples into a replay buffer. 2) With both the uploaded samples and the samples randomly sampled from the replay buffer, CEMA adapts the edge model $g_w(\cdot)$ with the guidance from the foundation model $f_\theta(\cdot)$ via the knowledge distillation loss (Eqn. 6).

lance camera in an industrial park) for delay-sensitive applications. In the inference, $g_w(\cdot)$ on edge devices may encounter out-of-distribution test samples. These test samples are distribution-shifted to the training ones due to natural variations or corruptions, such as lighting/weather changes and unexpected noises resulting from sensor degradation. In this case, the model $g_w(\cdot)$ may often be sensitive to these distribution shifts, potentially leading to significant performance degradation.

To tackle the distribution-shift issue, many test-time adaptation (TTA) approaches (Sun et al., 2020; Wang et al., 2021) have been proposed to improve the model adaptation performance through parameter updates. These methods become attractive since they do not require access to training data and adapt the model on the unlabeled test data via entropy minimization in a self-supervised manner. However, the applicability of these approaches to edge devices is limited due to computational resource constraints, such as memory limitations that hinder model updating. An alternative approach is to employ these methods to adapt the model $g_w(\cdot)$ in the cloud by centralizing the test data. Nevertheless, this suffers from two challenges. 1) Uploading all test samples incurs significantly heavy communication overhead. 2) Conventional TTA methods are typically designed for a single device and may be hard to fully exploit the resources of both the cloud and the edges. In this study, we seek to address the issues by developing an efficient and effective cloud-edge based adaptation method.

## 2.1 EFFICIENT ADAPTATION FOR ROBUSTNESS AND COMMUNICATION ENHANCEMENT

In this paper, we devise a Cloud-Edge Elastic Model Adaptation (CEMA) paradigm in which the adaptation task is decomposed to the cloud and edges based on their computational resources and budgets. The edge only requires performing a vanilla model inference, while the remaining adaptation workloads are offloaded to the cloud (see Figure 2). Then, our CEMA selectively uploads a subset of samples, determined by our proposed entropy-based criteria (refer to Section 2.2), to the cloud for adaptation. This selective sample uploading strategy significantly reduces the communication burden. Once the cloud adaptation process is complete, the edge model updates its parameters from the cloud and then infers the next incoming test samples. Importantly, CEMA introduces no extra computational cost in edges and is applicable to resource-constrained edge devices.

In the cloud, we seek to adapt the edge model with uploaded test samples. Specifically, we seek to leverage a foundation model $f_\theta(\cdot)$ with stronger capability and more parameters to guide the edge model for adaptation (refer to Section 2.3). Notably, the foundation model does not require access to training data and updates through unsupervised entropy minimization. To maximize data utilization for adaptation, we devise a replay buffer to store the uploaded samples. When transferring

| **Algorithm 1** Adaptation process in edge. | **Algorithm 2** Adaptation process in cloud. |
|---|---|

**Algorithm 1** Adaptation process in edge.

**Require:** Test samples $\mathcal{D}_{test}=\{\mathbf{x}_j\}_{j=1}^M$, the edge model $g_w(\cdot)$, parameters $B$, $E_{\max}$ and $E_{\min}$.

1: **for** a batch $\mathcal{X}=\{\mathbf{x}_b\}_{b=1}^B$ in $\mathcal{D}_{test}$ **do**
2:     Calculate predictions $\hat{y}$ for all $\mathbf{x} \in \mathcal{X}$ via $f_\Theta(\cdot)$.
3:     Calculate $S(\mathbf{x})$ via Eqn. (4) with $E_{\max}$ and $E_{\min}$.
4:     Update the threshold $E_{\max}$ via Eqn. (2).
5:     Upload samples $\{\mathbf{x}|S(\mathbf{x})=1, \mathbf{x}\in\mathcal{X}\}$ to cloud.
6:     Update the parameters $\tilde{w} \in w$ from the cloud.
7: **end for**
**Ensure:** The predictions $\{\hat{y}\}_{k=1}^M$ for all $\mathbf{x} \in \mathcal{D}_{test}$.

**Algorithm 2** Adaptation process in cloud.

**Require:** Test samples $\hat{\mathcal{X}}=\{\mathbf{x}_n\}_{n=1}^N$, the foundation model $f_\theta(\cdot)$ and edge model $g_w(\cdot)$.

1: Update parameters $\tilde{\theta} \in \theta$ of the foundation model $f_\theta(\cdot)$ via entropy minimization (Eqn. 5) with $\hat{\mathcal{X}}$.
2: Update parameters $\tilde{w} \in w$ of the edge model $g_w(\cdot)$ via knowledge distillation (Eqn. 6) with $\mathcal{X}$.
3: Distribute the parameters $\tilde{w}$ to edge.

knowledge from the foundation model to the edge model via knowledge distillation, we exploit both the newly uploaded samples and the samples from the replay buffer for higher data utilization efficiency. This results in better performance of the edge model compared to vanilla adaptation. The pseudo-code involved in the edge and cloud is presented in Algorithms 1 and 2, respectively.

## 2.2 SAMPLE FILTRATION FOR COMMUNICATION COST REDUCTION IN EDGE SIDE

To reduce the communication overhead between the cloud and edges, we propose a sample filtration strategy that removes high and low entropy test samples from being uploaded to the cloud. A recent study by Wang et al. (2021) proposes a model adaptation method that adapts on a batch of test samples by conducting entropy minimization. Minimizing entropy penalizes decisions at high densities in the data distribution to improve accuracy for distinct classes (Grandvalet & Bengio, 2004), which has proven to be a crucial constraint for domain adaptation (Saito et al., 2019; Roy et al., 2019). Furthermore, Niu et al. (2022) has demonstrated that high entropy samples adversely affect the adaptation performance when entropy minimization is employed. The reason may be that the model adapts using the test samples without labels via entropy minimization, introducing considerable uncertainty when dealing with high-entropy samples during the adaptation process.

However, this method only filters test samples based on a static and pre-determined threshold. It suffers two limitations: 1) The entropy of samples tends to decrease along with the adaptation. Therefore, only a part of the negatively impacting samples can be excluded. 2) It overlooks the fact that adaptation with extremely low-entropy samples is unnecessary. To address the above issues, we propose to 1) dynamically exclude the **unreliable** (high entropy) samples by adaptively adjusting the threshold in accordance with the entropy of current samples. 2) exclude the **low-informative** (low entropy) samples. We design the entropy-based filtration criteria as it is an information-theoretic measure that represents uncertainty and information and has proven to be a simple yet effective strategy (Settles, 2010; Margatina et al., 2021; Ren et al., 2022). To this end, we devise a binary score $S(\mathbf{x})$ to indicate whether a sample $\mathbf{x}$ should be uploaded. We only upload the test samples with $S(\mathbf{x})=\mathbf{1}$ and discard those with $S(\mathbf{x})=\mathbf{0}$.

**Dynamic identification on unreliable samples**. Let $\mathbb{1}_{\{\cdot\}}(\cdot)$ denote an indicator function. Following (Niu et al., 2022), we exploit a entropy threshold $E_{\max}$ to filter out the high entropy test samples as follows

$$S^{high}(\mathbf{x}) = \mathbb{1}_{\{E(\mathbf{x};w)<E_{\max}\}}(\mathbf{x}), \qquad (1)$$

where $E(\mathbf{x}; w)$ denotes the entropy of the prediction $g_w(\mathbf{x})$ for the sample $\mathbf{x}$. As we perform adaptation through entropy minimization, the entropy of the test samples is likely to decrease. Consequently, a fixed threshold $E_{\max}$ would progressively exclude fewer and fewer samples during the adaptation. To substantiate this, we perform an empirical study to reveal the proportions of the test samples whose entropy is larger than a fixed $E_{\max}$ at different stages

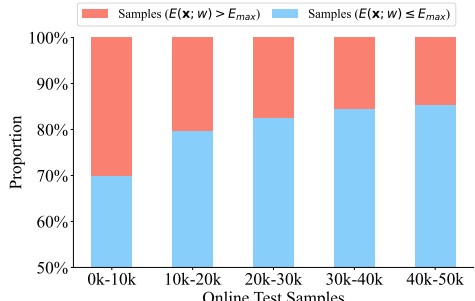

Figure 3: Proportions of test samples with $E(\mathbf{x}; w)>E_{\max}$ (red) and $E(\mathbf{x}; w)\leq E_{\max}$ (blue) during adaptation via entropy minimization on ImageNet-C.

during the adaptation process. As shown in Figure 3, in the final stage, we filter out fewer samples (indicated by the red bar) compared to the initial stage. This trend could be attributed to the predictions becoming more certain (*i.e.*, predictions with lower entropy) as the adaptation progresses.

In light of the aforementioned empirical study, it becomes feasible to filter out more high-entropy samples by dynamically decreasing the $E_{\max}$. To this end, we seek to lower $E_{\max}$ according to the average entropy of the test samples after every adaptation batch. To be specific, in the adaptation batch $t$, the entropy threshold $E_{\max}^t$ can be calculated by

$$E_{\max}^t \leftarrow \lambda \times E_{\max}^{t-1} \times \frac{E_{\mathrm{avg}}^t}{E_{\mathrm{avg}}^{t-1}}, \tag{2}$$

where $E_{\mathrm{avg}}^t$ denotes the average entropy of all test samples in past $t$ batches, $\lambda$ is a hyper-parameters. Note that $E_{\mathrm{avg}}^0$ can be obtained from the first batch of the test samples. Based on Eqn. (2), when the average entropy becomes smaller, $E_{\max}$ would be descended accordingly. In this way, we exclude more unreliable samples from uploading to the cloud, thereby enhancing communication efficiency.

**Identification on low-informative samples**. In addition to high entropy samples, test samples with extremely low entropy are unnecessary for adaptation. Since they only contribute negligible gradient while minimizing the entropy loss. Following the similar scheme above, we employ a threshold $E_{\min}$ to discard samples with entropy lower than $E_{\min}$. We do not adopt a dynamic variation strategy on $E_{\min}$ since the threshold that determines whether a sample contributes negligible gradient does not depend on the average entropy of current test samples. Formally, $S^{low}(\mathbf{x})$ can be written as

$$S^{low}(\mathbf{x}) = \mathbb{1}_{\{E(\mathbf{x};\theta) > E_{\min}\}}(\mathbf{x}). \tag{3}$$

The overall binary score $S(\mathbf{x})$ can be calculated by

$$S(\mathbf{x}) = S^{high}(\mathbf{x}) \cdot S^{low}(\mathbf{x}). \tag{4}$$

Note that the edge model only requires one regular forward propagation without the need for backward propagation or gradient descent. Once the edge model $g_w(\cdot)$ makes predictions for test samples, it asynchronously uploads the samples with $S(\mathbf{x}){=}1$ to the cloud using a background thread. We emphasize that uploading samples does not block the edge from inferring on next incoming samples. In other words, the processes of inference and uploading can be executed simultaneously.

## 2.3 Replay-based Knowledge Distillation for Adaptation in Cloud Side

Recent studies have demonstrated that larger models with a great number of parameters often achieve better performance than a small one on the out-of-distribution data (Hendrycks & Dietterich, 2019). Compared with the vanilla adaptation, it is possible to further improve the edge model $g_w(\cdot)$ in the cloud by distillation from a high-performing foundation model $f_\theta(\cdot)$. Note that it is feasible and practical that the foundation model has knowledge that covers all the test samples inferred by the edge model (see results and analysis in Table 8). Since cloud server has much more sufficient computational resources and budgets to support the heavier models. In this case, our proposed CEMA would adopt a foundation model to enhance the adaptation performance, which takes advantage of the rich computational resources of the cloud.

However, it is non-trivial to distill the foundation model $f_\theta(\cdot)$ to the edge model $g_w(\cdot)$. Note that vanilla distillation training requires a large number of samples (Chen et al., 2019; Yu et al., 2023). The amount of accessible test samples is limited as we exclude unreliable and low-informative samples in the edge. To alleviate this issue, we devise a replay buffer $\mathcal{B}$ to store the uploaded samples inspired by (Mnih et al., 2013). In each distillation step, we first update the foundation model $f_\theta(\cdot)$ via unsupervised entropy minimization in an unsupervised manner on the uploaded samples. Through this adaptation process, the foundation model acquires additional knowledge of the current test data. Then, we boost the edge model $g_w(\cdot)$ via the knowledge distillation guided by the foundation model on both the uploaded samples and the samples from the replay buffer.

**Entropy minimization for foundation model adaptation**. Upon receiving the batch of uploaded test samples $\hat{\mathcal{X}}{=}\{\mathbf{x}_i\}_{i=1}^N$, we first put them into the replay buffer $\mathcal{B}{=}\mathcal{B} \cup \hat{\mathcal{X}}$. Note that the capacity

of the buffer is limited (see analysis and ablations in Table 16). We update the buffer with the newly uploaded test samples in a first-in and first-out manner. Subsequently, we update the foundation model over the batch $\hat{\mathcal{X}}$. Adaptation with a batch not only avoids a trivial solution (*i.e.*, assigning all probability to the most probable class (Wang et al., 2021)) but also improves the parallel efficiency in the cloud server. Formally, we adapt $f_\theta(\cdot)$ by minimizing the weighted entropy loss $\mathcal{L}_{\text{ENT}}(f_\theta(\mathbf{x}))$

$$\min_{\tilde{\theta}} H(\mathbf{x}) \sum_{y \in \mathcal{C}} f_\theta(y|\mathbf{x}) \log f_\theta(y|\mathbf{x}), \tag{5}$$

where $H(\mathbf{x}) = 1/\exp(E(\mathbf{x}; \theta) - E_{\max})$ and $\mathcal{C}$ denotes the model output space. In Eqn. (5), we optimize the loss weighted by $H(\mathbf{x})$ following (Niu et al., 2022) since we seek to encourage the model to focus on the low-entropy test samples during adaptation. In the back-propagation process, it is worth noting that the gradient is not propagated through $H(\mathbf{x})$.

For efficient adaptation, we only update the affine parameters of normalization layers $\tilde{\theta} \in \theta$ while keeping the remaining layers fixed. The advantages are three folds: 1) This model updating strategy is efficient since it requires less memory footprint and computational resources. 2) Compared with distributing all the parameters to the edge, our CEMA only needs to transfer BN parameters and lowers downloading overhead (*e.g.*, reduce 99.91% parameters distributing burden in ResNet-18). 3) The model may preserve the previous knowledge since most parameters remain unchanged, leading to better adaptation stability and performance (see results in Table 17).

**Knowledge distillation with replay buffer for edge model adaptation**. With the updated foundation model, we seek to adapt the edge model $g_w(\cdot)$ on both $\hat{\mathcal{X}}$ and a set of test samples randomly sampled from the replay buffer $\mathcal{B}$. Specifically, we align the predictions between the logits $f_\theta(\cdot)$ and $g_w(\cdot)$ via Kullback–Leibler (KL) divergence $\mathcal{L}_{\text{KL}}$. In addition, we adapt $g_w(\cdot)$ via a cross-entropy loss $\mathcal{L}_{\text{CE}}$, in which the pseudo labels $\hat{y}$ are generated by the foundation model. Note that $\hat{y}$ can be calculated by $\hat{y} = \arg\max_{y \in \mathcal{C}}(f_\theta(y|x))$. Formally, we optimize $g_w(\cdot)$ by employing both entropy minimization and knowledge distillation as follows,

$$\min_{\tilde{w}} H(\mathbf{x})[\alpha \mathcal{L}_{\text{KL}}(g_w(\mathbf{x}), f_\theta(\mathbf{x})) + \beta \mathcal{L}_{\text{CE}}(g_w(\mathbf{x}), \hat{y}) + \mathcal{L}_{\text{ENT}}(g_w(\mathbf{x})))], \tag{6}$$

where $\alpha$ and $\beta$ are factors for balancing the losses. we use the KL divergence to align the prediction distribution of the foundation and edge models, and the CE loss to align the decision boundaries. These two kinds of losses make the edge model learn the knowledge in complementary manners from the foundation model. The combination of both losses outperforms the adoption of either one in isolation (See results and analyses in Table 12).

In edge model adaptation, we still only update the affine parameters of normalization layers $\tilde{w} \in w$. After distilled adaptation of the edge model $f_w(\cdot)$, we distribute the parameters $\tilde{w}$ to edge devices to update $f_w(\cdot)$. In particular, once the cloud has completed adaptation, we can perform the reception of parameters from the cloud in the background, typically via a separate thread to avoid blocking the inference process. Upon full reception of the parameters, we would update the parameters of the edge model at the beginning of the next inference iteration.

## 3 EXPERIMENTS

In the following, we provide the details of the used datasets and implementation, more details are put in the supplementary materials. The code is available at https://github.com/chenyaofo/CEMA.

**Datasets and models**. We evaluate our method and considered methods on ImageNet-C (Hendrycks & Dietterich, 2019). which is a distribution shift dataset by applying 4 main category corruption (*i.e.*, noise, blur, weather, and digital), with a total of 15 diverse corruption types, to the ImageNet validation set. Each corruption has 5 different severity levels (*i.e.*, from level 1 to 5), in which the higher severity level indicates the more severe distribution shift. We also verify our CEMA on ImageNet-R (Hendrycks et al., 2021), which contains 30,000 images with various artistic renditions of 200 ImageNet classes collected from Flickr. We adopt ResNet101 (He et al., 2016) as the foundation model and ResNet18 as the edge model in the main experiments. We explore more foundation models and edge models in the ablation studies (see results in Tables 13 and 14).

**Implementation details**. We set the entropy thresholds $E_{\max} = 0.4 \times \ln C$ as the initialized value following (Niu et al., 2022) and $E_{\min} = 0.02 \times \ln C$, where $C$ denotes the number of classes. Then

Table 1: Comparisons with state-of-the-art methods on ImageNet-C (severity level 3 and 5) regarding **Accuracy (%)**. We adopt Resnet101 as the foundation model and ResNet18 as the edge model. † denotes the TTA method that does not require any backward propagation and can be locally executed in edge devices. Note that our CEMA requires uploading fewer samples on average with severity levels 3 (19.1k *vs.* 26.1k) and 5 (14.4k *vs.* 18.8k) than ETA.

| | Noise | | | Blur | | | | Weather | | | | Digital | | | | |
|---|---|---|---|---|---|---|---|---|---|---|---|---|---|---|---|---|
| Severity Level=3 | Gauss. | Shot | Impul. | Defoc. | Glass | Motion | Zoom | Snow | Frost | Fog | Brit. | Contr. | Elastic | Pixel | JPEG | Avg. |
| ResNet18 (baseline) | 21.6 | 19.9 | 18.7 | 29.9 | 15.8 | 28.7 | 27.6 | 27.6 | 23.8 | 35.5 | 62.7 | 38.1 | 51.8 | 41.6 | 53.0 | 33.1 |
| • BN Adaptation† | 42.3 | 39.8 | 40.0 | 37.5 | 31.4 | 45.1 | 44.3 | 40.8 | 36.2 | 53.9 | 65.0 | 58.2 | 60.2 | 58.0 | 57.7 | 47.4 |
| • ONDA† | 40.0 | 38.9 | 37.5 | 29.5 | 27.5 | 43.8 | 43.9 | 40.2 | 35.2 | 54.6 | 65.1 | 56.1 | 59.7 | 58.6 | 57.6 | 45.9 |
| • LAME† | 20.6 | 18.9 | 17.2 | 29.5 | 14.7 | 28.3 | 26.9 | 26.8 | 23.2 | 34.9 | 62.4 | 37.5 | 51.3 | 41.1 | 52.5 | 32.4 |
| • PL | 48.1 | 48.0 | 46.1 | 41.1 | 39.7 | 51.3 | 49.9 | 47.3 | 39.8 | 58.6 | 64.9 | 59.2 | 62.5 | 60.8 | 59.4 | 51.8 |
| • Tent | 47.2 | 47.1 | 45.1 | 40.0 | 38.2 | 50.4 | 49.4 | 46.7 | 40.1 | 58.1 | 64.9 | 59.0 | 62.5 | 60.5 | 59.2 | 51.2 |
| • CoTTA | 42.0 | 40.7 | 39.8 | 30.3 | 30.1 | 46.3 | 46.1 | 41.9 | 36.5 | 56.2 | 64.9 | 58.0 | 60.2 | 59.3 | 58.1 | 47.4 |
| • ETA | 50.1 | 50.2 | 48.6 | 44.0 | 42.7 | 52.9 | 51.4 | 49.9 | 43.5 | 59.5 | **65.2** | 60.9 | **62.9** | 61.6 | **59.9** | 53.5 |
| • CEMA (Ours) | **51.1** | **51.2** | **49.8** | **45.2** | **44.1** | **53.7** | **52.0** | **50.8** | **44.2** | **60.1** | 65.0 | **61.1** | **62.9** | 61.6 | 59.8 | **54.2** |
| Severity Level=5 | Gauss. | Shot | Impul. | Defoc. | Glass | Motion | Zoom | Snow | Frost | Fog | Brit. | Contr. | Elastic | Pixel | JPEG | Avg. |
| ResNet18 (baseline) | 1.5 | 2.3 | 1.5 | 11.4 | 8.7 | 11.1 | 17.6 | 10.6 | 16.2 | 14.0 | 51.5 | 3.4 | 16.5 | 23.3 | 30.7 | 14.7 |
| • BN Adaptation† | 16.6 | 16.2 | 17.3 | 18.6 | 18.2 | 25.9 | 34.7 | 28.4 | 29.8 | 41.2 | 58.5 | 22.2 | 40.1 | 45.3 | 38.0 | 30.1 |
| • ONDA† | 13.7 | 15.0 | 14.1 | 12.3 | 13.2 | 23.7 | 34.2 | 29.4 | 28.6 | 40.9 | 58.5 | 12.3 | 39.3 | 44.6 | 37.5 | 27.8 |
| • LAME† | 0.9 | 1.1 | 0.6 | 11.2 | 8.2 | 10.8 | 17.0 | 8.7 | 15.6 | 12.4 | 51.1 | 3.3 | 14.9 | 22.5 | 30.1 | 13.9 |
| • PL | 24.8 | 26.8 | 24.6 | 20.3 | 21.3 | 33.6 | 41.8 | 39.0 | 32.4 | 49.9 | 59.5 | 11.4 | 47.9 | 51.5 | 47.0 | 35.4 |
| • Tent | 22.8 | 25.0 | 23.2 | 20.1 | 21.1 | 32.4 | 41.0 | 37.8 | 33.5 | 48.9 | 59.3 | 18.0 | 46.9 | 50.6 | 45.9 | 35.1 |
| • CoTTA | 15.2 | 16.2 | 15.7 | 11.8 | 14.9 | 26.9 | 36.9 | 31.2 | 29.9 | 43.6 | 59.2 | 17.0 | 40.9 | 47.2 | 39.3 | 29.7 |
| • ETA | 26.8 | 29.7 | 27.6 | 22.6 | 22.7 | 37.1 | 44.0 | 42.4 | 37.6 | 51.6 | 60.1 | 26.1 | 49.8 | 53.3 | 48.5 | 38.7 |
| • CEMA (Ours) | **29.8** | **32.2** | **30.3** | **25.3** | **26.8** | **39.3** | **45.3** | **43.7** | **38.7** | **52.8** | **60.1** | **32.9** | **50.8** | **54.0** | **49.3** | **40.8** |

the threshold $E_{\max}$ decreases based on Eqn. (2) with $\lambda=1.0$. For the adaptation of the foundation and edge model, we use both an SGD optimizer with a learning rate of 0.00025 and a momentum of 0.9. For the adaptation of the edge model, we set the batch size to 128, in which 32 samples are newly uploaded and the remaining 96 samples are randomly sampled from the replay buffer. The hyper-parameter $\alpha$ and $\beta$ are both set to 3.

**Compared methods**. We compare our methods with the following state-of-the-art TTA methods, including BN Adaptation (Schneider et al., 2020), ONDA (Mancini et al., 2018), LAME (Boudiaf et al., 2022), Pseudo Label (PL) (Lee et al., 2013), Tent (Wang et al., 2021), CoTTA (Wang et al., 2022) and ETA (Niu et al., 2022). In the experiments, we assume the edge devices only have limited resources and thus are unable to perform backpropagation. All workloads in the above TTA methods invoked in backpropagation would be executed in the cloud by uploading test samples. In this way, we can compare the amount of data transmission between our method and the counterparts.

3.1 PERFORMANCE COMPARISONS ON IMAGENET-C

We compared our proposed CEMA with the considered methods in Table 1 in ImageNet-C (Hendrycks & Dietterich, 2019) with the severity levels 3 and 5. We adopt a CNN-based model ResNet101 as the foundation model and ResNet18 as the edge model in this experiment. We put more experimental results with the severity levels 1, 2, and 4 as well as the results on transformer-based models in the supplementary due to the page limitation. From the results, our CEMA achieves the highest accuracy in most 15 different corruption types and the best average accuracy (*e.g.*, 40.8% with the severity level 5). To be specific, CEMA outperforms Tent (29.8% *vs.* 22.8%) and ETA (29.8% *vs.* 26.8%) on the corruption Gaussian noise with the severity level 5. The reasons are twofold: 1) the proposed filtration strategy removes more harmful test samples; 2) our replay-based distillation scheme transfers distribution knowledge to the edge model. The results verify the effectiveness of the proposed sample filtration strategy and distilled adaptation method.

In Figure 4, we compare the average number of uploaded samples over 15 corruption types of our CEMA and the considered methods with ResNet18 as the edge model. Note that the compared methods PL, Tent and CoTTA need to upload the whole test samples (50k samples, 100%) to the cloud for adaptation. These methods introduce great communication overhead between the cloud and edges, which is inefficient in the context of cloud edge model deployment. ETA excludes some test samples with high prediction entropy but still requires uploading 26.1k (52%) and 19.1k (38%) samples in severity levels 3 and 5, respectively. With the proposed dynamic sample filtration

Table 2: Comparisons with state-of-the-art methods on ImageNet-R benchmark. We adopt ResNet101 as the foundation model and ResNet18 as the edge model. $^\dagger$ denotes the TTA method that does not require any backward propagation and can be executed in edge devices, which does not require uploading test samples.

| Model | Acc. (%) | #Uploaded samples |
|---|---|---|
| ResNet18 (baseline) | 20.4 | – |
| • BN Adapt.$^\dagger$ | 22.8 | – |
| • ONDA$^\dagger$ | 22.7 | – |
| • LAME$^\dagger$ | 20.2 | – |
| • PL | 23.8 | 30,000 (100%) |
| • Tent | 23.6 | 30,000 (100%) |
| • ETA | 26.2 | 7,457 (25%) |
| • CoTTA | 23.2 | 30,000 (100%) |
| • CEMA (Ours) | **27.9** | **5,944 (20%)** |

Table 3: Comparisons with Tent and ETA with a mixture of 15 corruption types on ImageNet-C.

| Model | Accuracy (%) | #Uploaded samples |
|---|---|---|
| ResNet18 (baseline) | 33.1 | – |
| • Tent | 30.5 | 750k (100%) |
| • ETA | 44.5 | 384k (51%) |
| • CEMA (Ours) | **45.6** | **286k (38%)** |

Table 4: Effect of sample filtration strategy.

| Method | Level 3 Acc. (%) | Level 3 #Upload | Level 5 Acc. (%) | Level 5 #Upload |
|---|---|---|---|---|
| $S^{high}(\mathbf{x})$ w/ Fixed $E_{max}$ | 51.2 | 25,325 | 30.0 | 15,473 |
| + Dynamic $E_{max}$ | 51.1 | 20,976 | 29.9 | 10,081 |
| + $S^{low}(\mathbf{x})$ | **51.1** | **17,479** | **29.8** | **9,889** |

Table 5: Effect of the replay buffer.

| Replay Buffer | Level 3 Acc. (%) | Level 3 #Upload | Level 5 Acc. (%) | Level 5 #Upload |
|---|---|---|---|---|
| × | 47.4 | 17,654 | 25.0 | 9,084 |
| ✓ | **51.1** | 17,479 | **29.8** | 9,889 |

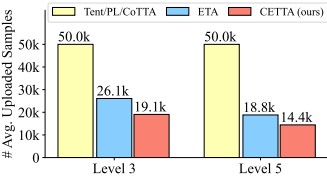
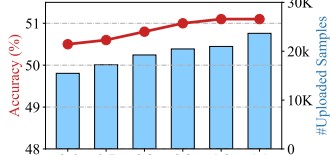
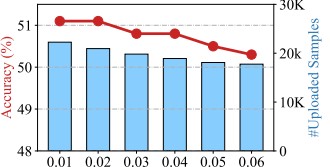

Figure 4: Comparisons of the average number of uploaded test samples on ImageNet-C with the severity levels 3 and 5.

Figure 5: Effect of $\lambda$ in Eqn. (2) with ResNet18 as edge model on ImageNet-C (Gaussian noise, severity level 3).

Figure 6: Effect of $E_{min}$ in Eqn. (3) with ResNet18 as edge model on ImageNet-C (Gaussian noise, severity level 3).

strategy, our CEMA further reduces the number of uploaded samples to 18.8k (37%) and 14.4k (29%) in severity levels 3 and 5. Compared with ETA, we remove more high-entropy samples with a dynamic threshold and further exclude low-entropy samples. In this case, our proposed CEMA only needs to upload fewer samples to the cloud for adaptation, which demonstrates the superiority of our methods over existing methods in terms of the amount of data transmission.

### 3.2 PERFORMANCE COMPARISONS ON IMAGENET-R

In Table 2, we report the results of our CEMA and considered methods on a realistic out-of-distribution dataset ImageNet-R. This benchmark dataset is collected from Flickr instead of generated by some corruption algorithms (*i.e.*, as ImageNet-C done) and filtered by Amazon MTurk annotators according to the class names in the original ImageNet. From the results, our CEMA achieves 27.9% accuracy on ImageNet-R (+4.1% over Tent, +1.7% over the best counterpart ETA). As for the number of uploaded test samples, our CEMA only requires 5944 (20%) samples, which is lower than ETA (7457, 25%) and much lower than Tent, PL and CoTTA (30,000, 100%). These results are consistent with those on ImageNet-C that our proposed CEMA yields the highest robust accuracy with the fewest uploaded samples. The results further demonstrate the effectiveness and potential of our method while applied to realistic test samples in real-world applications.

### 3.3 FURTHER EXPERIMENTS

In ablation studies, we consider two representative severity levels (*i.e.*, 3 and 5). Severity levels 3 and 5 represent the medium-difficulty distribution shift and the most challenging distribution shift, respectively. By adopting these two levels, we effectively assess the performance of various adaptation algorithms. Moreover, the hyper-parameters derived from these levels demonstrate their adaptability

and suitability for addressing other severity levels as well. For a similar consideration, the settings are widely adopted by Tent (Wang et al., 2021) and ETA(Niu et al., 2022).

Due to the page limitation, We put more ablations in the supplementary materials, including the effects of 1) hyperparameters $E_{\max}$, $\alpha$ and $\beta$, 2) components in distillation loss (Eqn. (6)), 3) different foundation models, 4) different edge models, 5) different filtration strategies 6) replay buffer size, 7) different parameters updating schemes and 8) different parameters distribution intervals.

**Comparisons under mixed-and-shifted distributions.** We evaluate our CEMA on mixed ImageNet-C in severity level 3 that consists of 15 different corruption types/distribution shifts (750k images in total). Note that it is common for the edge model to encounter test samples with mixed distribution shifts in practice. In Table 3, compared with Tent and ETA, our CEMA outperforms them in both the accuracy (+15.1% over Tent and +1.1% over ETA) and the number of uploaded samples (-62% than Tent, -13% than ETA). The results show the effectiveness of CEMA for applications in complex scenarios with out-of-distribution samples.

**Effect of $\lambda$ in Eqn. (2).** To investigate the effect of $\lambda$ in Eqn. (2), we perform more experiments with different $\lambda \in \{0.6, 0.7, 0.8, 0.9, 1.0, 1.1\}$. From Figure 5, when $\lambda$ becomes larger, our CEMA would remove fewer test samples and upload more samples to the cloud, and the robust accuracy improves since we transfer the more contributed samples to the cloud for adaptation. The robust accuracy is highest (51.1%) when $\lambda$=1.0 and keep unchanged while $\lambda$>1.0. Considering a larger $\lambda$ leads to more communication burden, we select $\lambda$=1.0 for the efficiency-performance trade-off and fix $\lambda$ to 1.0 for all other experiments. Experimental results in Tables 1, and 2 demonstrate that this fixed $\lambda$ works well, indicating that the $\lambda$ in our CEMA is not sensitive to different datasets.

**Effect of $E_{\min}$ in Eqn. (3).** We evaluate our CEMA with different $E_{\min}$, selected from $\{0.01, 0.02, 0.03, 0.04, 0.05, 0.06\}$. Note that the larger $E_{\min}$ is, the more low-entropy samples are excluded. In this case, the out-of-distribution performance would decrease since we may remove some contributed samples during adaptation. From Figure 6, the accuracy drops when $E_{\min}$ is larger than 0.02. Therefore, we set the entropy threshold $E_{\min}$ to 0.02 across various datasets, including ImageNet-C (15 corruption types and 5 severity levels) and ImageNet-R. Extensive experiments show that our CEMA works well with the chosen hyperparameters on different datasets and various severity levels.

**Effect of components in sample filtration strategy.** We perform an ablation experiment in Table 4 to verify the components in the proposed sample filtration strategy. Compared with the baseline that removes test samples based on $S^{high}(\mathbf{x})$ with fixed $E_{\max}$ (the same as ETA), introducing dynamic $E_{\max}$ in Eqn. (2) achieves comparable accuracy (51.1% *vs.* 51.2%) in severity level 3 with fewer uploaded samples (20,976 *vs.* 25,325). When further removing low-entropy samples based on $S^{low}(\mathbf{x})$ in Eqn. (3), the number of uploaded samples further decreases (*e.g.*, 25,325 → 17,479). The results demonstrate the effectiveness of our proposed sample filtration strategy. We further demonstrate the superiority of our filtration strategy over the random uploading strategy in Table 15.

**Effect of the replay buffer.** In Table 5, we report the performance of our CEMA on ImageNet-C (Gaussian noise, severity level 3) with/without the replay buffer. Note that the replay buffer is able to provide more samples for distillation and boost the data utilization efficiency in our cloud edge adaptation. With the replay buffer, our CEMA achieves much higher accuracy (51.1% *vs.* 47.4%) on ImageNet-C (Gaussian noise, severity level 3). The reason is that we reuse samples from the replay buffer for knowledge distillation. This leads to more sufficient model updates when the foundation model transfers the knowledge to the edge model. The experimental results demonstrate the effectiveness of the proposed replay buffer in knowledge distillation.

# 4 CONCLUSION

In this paper, we devise a Cloud-Edge Elastic Model Adaptation (CEMA) paradigm that improves the model adaptation performance by leveraging both the cloud server and edge devices. In the adaptation, we highlight that our CEMA does not introduce any extra computational cost in the edge devices. Specifically, we devise a sample filtration strategy to exclude unnecessary samples from the cloud for adaptation. It reduces the data transmission overhead between the cloud and edge and thus improves the adaptation efficiency. Besides, we adopt a powerful foundation model to guide the edge model for adaptation via the proposed replay-based knowledge distillation. Extensive experimental results on several benchmark datasets demonstrate the effectiveness of our CEMA.

## ACKNOWLEDGEMENTS

This work was partially supported by National Natural Science Foundation of China (NSFC) 61836003 (key project), National Natural Science Foundation of China (NSFC) 62072190, the Major Key Project of PCL PCL2023A08 and TCL Science and Technology Innovation Fund.

## REPRODUCIBILITY STATEMENT

In this work, we implement our CEMA with different models on ImageNet-C and ImageNet-R datasets. Reproducing all the results in our method depends on the following three aspects:

1. **DATASET.** The second paragraph of Section 3 and Appendix C.1 provide the details of the adopted dataset and the download `url`.

2. **MODELS.** All adopted models (with the pre-trained weights) for test-time adaptation are publicly available. The download `url` is provided in Appendix C.2.

3. **PROTOCOLS OF EACH METHOD.** Appendix C.2 provides the implementation details of our CEMA and compared methods. The source code of our CEMA has been made publicly available.

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

# APPENDIX

## CONTENTS

# A  RELATED WORK

**Test-time adaptation (TTA)** (Sun et al., 2020; Wang et al., 2021) recently has shown great potential in handling distribution shifts between training and testing data, by directly adapting the pre-trained model on a given test sample to learn the shifts. Specifically, test-time training (Sun et al., 2020) updates the model at test time via a self-supervised image rotating prediction (Gidaris et al., 2018). After that, numerous methods (Zhang et al., 2022b; Wang et al., 2022; Bartler et al., 2022; Niu et al., 2022; 2023; Wen et al., 2023) have been further devised to improve and broaden the application scope of TTA. For example, TTT++ (Liu et al., 2021) and MT3 (Bartler et al., 2022) introduce self-supervised contrastive learning (Chen et al., 2020b) to provide supervision for model updating, which achieves better adaptation performance. Meanwhile, entropy-based methods (Wang et al., 2021; Niu et al., 2022; 2023; Zhang et al., 2022a) update the model by minimizing the unsupervised entropy of model outputs. However, the above methods rely on backpropagation at test time, which may be infeasible for resource-limited edge devices. To avoid this issue, one can update the model via a backpropagation-free manner, such as BN statistics adaptation (Schneider et al., 2020; Hu et al., 2021; Nado et al., 2020; Khurana et al., 2021), classifier adjustment (Iwasawa & Matsuo, 2021) and reconstruction learning (Gandelsman et al., 2022; Deng et al., 2024). Though these methods improve the efficiency of TTA, they may yield inferior performance due to the insufficient model update.

In this work, we focus on deploying backpropagation-based model adaptation to cloud-edge systems to boost model adaptation performance. Considering the computational resources of the cloud and edges, our method only performs forward propagation without model updating on edge devices and does not introduce any extra computational cost. Instead, we adapt the model in the cloud by uploading only partial test samples from the edges. In this way, we allocate all the heavy test-time adaptation workloads to the cloud and reduce the computational cost of edges.

**Knowledge distillation**. Knowledge distillation (Hinton et al., 2015) is an effective method to obtain simple and efficient student models by transferring knowledge from complex teacher models. According to the types of extracted knowledge, knowledge distillation can be divided into logits-based distillation methods (Hinton et al., 2015; Yang et al., 2022a; Zhao et al., 2022), feature-based distillation methods (Romero et al., 2015; Chen et al., 2021a;b; Yang et al., 2022b) and relation-based distillation methods (Park et al., 2019; Liu et al., 2019; Ye et al., 2022). The above methods are mostly based on the pre-trained teacher models for offline distillation. Related to our method, online distillation methods (Zhang et al., 2018; Chen et al., 2020a; Bhat et al., 2021; Qian et al., 2022) can train teacher models and student models simultaneously in the absence of strong teachers. In this paper, we exploit knowledge distillation to leverage a stronger foundation model in the cloud to boost the robustness of an edge model in the context of cloud-edge inference.

**Collaborative cloud-edge inference**. Conventional cloud-based model inference (Olston et al., 2017; Vanholder, 2016; Crankshaw et al., 2017) has an unacceptable responsive latency concern, which limits its applications in practice. To overcome these shortcomings, collaborative cloud-edge inference (Osia et al., 2018; Wang et al., 2018; Xue et al., 2021; Ren et al., 2021; Liu et al., 2020) leverages both the computational power on the edge and the cloud by dynamically allocating the workloads on them, which introduces plenty of advantages (Zhou et al., 2019), such as low response latency and on-demand deployment. These methods mostly focus on the vanilla model inference task, *i.e.*, a model simply takes a test instance as input and outputs its predictions. Unlike these methods, we seek to improve the robustness in cloud-edge inference via test-time adaptation to alleviate the distribution shift issue. To this end, we propose to divide the TTA task into several subtasks (including edge model inference and adaptation) and offload them efficiently between the cloud and edges.

Related to our CEMA, DUET (Lv et al., 2023) and HyperNetwork-based method (Alanov et al., 2022) seek to generate model parameters in the test time to address the distribution-shifted issue with cloud-edge collaboration. However, DUET requires uploading all test samples to update the parameter generator. While our CEMA excludes unnecessary samples from uploading to the cloud. Thus our CEMA may be applied to bandwidth-limited scenarios. Moreover, DUET updates the parameter generator in a supervised way, which requires source training data and corresponding ground-truth labels. While our CEMA performs model adaptation in a self-supervised manner only with unlabeled samples. Our CEMA may be applicable to more practical scenarios. The hyperNetwork-based method requires a prior specification of domain shift categories (*i.e.*, represented by texts) when

training its HyperDomainNet. While in our CEMA, the domain shifts are unknown in the training stage. Consequently, the application of HyperDomainNet to our task presents inherent difficulties.

**Informative sample identification** seeks to quantitatively measure the information contained in a given sample and then develop sample-aware learning strategies, which has proven to be effective in many areas, such as active learning (Ren et al., 2022; Gal et al., 2017; Sener & Savarese, 2018; Beluch et al., 2018; Tsymbalov et al., 2018) and domain adaptation (Zhang et al., 2020; Prabhu et al., 2021). For instance, BLAD (Gal et al., 2017) measures the sample's information by estimating the mutual information between model parameters and model predictions, and dropout-based methods (Tsymbalov et al., 2018) calculate the variation over a set of model predictions with dropout to quantify the information. Sener & Savarese (2018) propose a core-set approach to select samples that can mostly represent the whole dataset. SENTRY (Prabhu et al., 2021) and CoUDA (Zhang et al., 2020) measure the sample information via the prediction consistency of different data augmentations and networks, respectively.

In our CEMA, one key challenge is to reduce the data transmission costs in the context of efficient online test-time cloud-edge model adaptation. We achieve this by employing the idea of sample selection. To be specific, we devise an entropy-based thresholding technique to exclude partial samples from the model adaptation process. Here, we adopt the entropy as the sample's information measurement, as it is easy to use and efficient. The entropy can be calculated over a single sample and only involves one-time forward propagation, unlike prior methods that may rely on a whole dataset (Sener & Savarese, 2018) or less-efficient multiple forward propagations (Tsymbalov et al., 2018; Prabhu et al., 2021). Nonetheless, compared with EATA (Niu et al., 2022) in which the authors exploit a static thresholding strategy to select unreliable samples, our sample selection strategy is different in the two aspects: i) we reveal that the suitable threshold may change continuously along with the online adaptation process and propose a dynamic unreliable sample identification strategy; ii) we also introduce a low-informative sample selection strategy to identify samples that produce negligible gradients for model updating.

**Self-paced learning**. Motivated by the learning principle of humans, self-paced learning (Kumar et al., 2010) automatically reorders samples during training based on their difficulty. For instance, SPL (Kumar et al., 2010) and SP-MIL (Sangineto et al., 2019) iteratively select a subset of the most reliable images for model updates. To enhance the diversity of the selected samples, SPLD (Jiang et al., 2014) pre-cluster the training data and encourages balance samples section from different clusters. Furthermore, SP-CON (Peng et al., 2021) jointly learns the important weights for each sample during training. Based on this, they conduct self-paced learning with the importance weights incorporated within the loss. Despite both self-paced learning and CEMA improving learning efficiency by active sample selection, self-paced learning focuses on learning robustness instead of mitigating computation cost. Consequently, it differs from CEMA in several aspects: 1) Self-paced learning initiates the learning process with an easy subset of samples, which includes even those of low informativeness. In contrast, our CEMA approach specifically focuses on selecting samples that are both informative and reliable for adaptation. These two kinds of samples are beneficial to the adaptation process. 2) Self-paced learning offline selects the whole dataset for training as the remaining samples become easier. While our CEMA dynamically adjusts the threshold to continually filter out less reliable samples. This is conducted online, focusing on maintaining communication efficiency.

## B MORE DISCUSSIONS ON CEMA

### B.1 TRANSMISSION EFFICIENCY OF CEMA

We would like to highlight that our CEMA reduces the uploading and downloading communication cost from two aspects. 1) **Reducing uploading communication cost**: We design entropy-based criteria to exclude unreliable and low-informative samples. It reduces 60% uploading communication overhead on ImageNet-C (Gaussian noise, severity level 3) benchmark. 2) **Reducing downloading communication cost**: We only update and transfer the affine parameters in BN layers instead of all the layers. Compared with the distribution of all the parameters in ResNet18 (11.68 MB), our CEMA only needs to transfer 0.0096 MB parameters and lowers 99.91% downloading communication overhead.

### B.2 ADAPTATION THROUGHPUT AND REQUIRED UPLOAD BANDWIDTH

Taking ResNet101 as the foundation model and ResNet18 as the edge model, our CEMA can run at 220 images/second on NVIDIA A100 GPU. This is sufficient to support edge devices for real-time inference (60 images/second). On the edge side, we assume the edge model infers at 60 images/second. Based on that our CEMA reduces 60% uploading communication overhead on ImageNet-C (Gaussian noise, severity level 3), it needs to upload around 24 images to the cloud per second. Each image on ImageNet-C is around 28 KB in disk after jpeg compression. In total, the edge device needs to upload around 24×28=672 KB data to the cloud per second.

### B.3 AVAILABILITY IN VARIABLE BANDWIDTH SCENARIOS

In practice, the bandwidth between the cloud and the edge may change constantly. Nevertheless, in certain contexts, such as surveillance systems in industrial parks, bandwidth is not a major concern since 1) our CEMA only requires relatively low communication overhead and 2) the cloud and edge devices are commonly interconnected through high-speed wired networks (typically with 100Mb/s bandwidth). Introducing the bandwidth variable in our CEMA method would significantly increase the complexity, which may make the adaptation performance unstable.

We intend to address this issue by exploiting an uploading queue. Specifically, we will upload test samples with the queue in the background thread, while ensuring that it does not block foreground model inference tasks. When the queue reaches its maximum capacity, we will discard the test sample with the highest entropy in the queue. Through this mechanism, our CEMA is able to dynamically adjust the number of uploaded samples according to the available bandwidth. In this sense, it is equivalent to adjusting the hyperparameter $\lambda$ in Eqn. (2) according to the available bandwidth instead of manually pre-defining it. Thus, our proposed method is well-suited to handle diverse and varying bandwidth conditions.

### B.4 ADAPTATION AND PARAMETER UPDATING MECHANISMS

In our CEMA, we feed a sample into the edge model and then determine whether this sample would be uploaded to the cloud based on Eqn. (4). Once receiving a batch of $N$ uploaded samples in the cloud, the edge model would be adapted for one time via Eqn. (6). After adaptation, the edge model would update the parameters from the cloud. Thus the next coming sample would be inferred via the updated edge model. We have also taken into account scenarios of poor network connectivity (as detailed in Table 18). In this case, the edge model would be adapted for $K$ time ($K > 1$) via Eqn. (6). Then the updated parameters would be distributed to the edge and the next coming sample would be inferred via the updated edge model.

### B.5 ENTROPY-BASED CRITERION ON OVERCONFIDENCE MODELS

Recent studies have demonstrated that neural networks can exhibit overconfidence, which may have an influence on the measure of uncertainty based on entropy-based criteria. To evaluate the overconfidence in our edge model, we employ a commonly used metric Expected Calibration Error (ECE) (Naeini et al., 2015). ECE measures the average differences between the model's predicted

confidence and its actual accuracy across various confidence intervals. A lower ECE value indicates reduced overconfidence in the model's predictions. Compared with the best counterpart ETA (ECE=6.67%), our CEMA achieves ECE=3.07%. This substantial reduction in ECE suggests that the edge model in CEMA exhibits significantly less overconfidence. Consequently, the uncertainty estimation using entropy in CEMA could potentially be more accurate than that in ETA.

Moreover, it is important to acknowledge that a certain degree of overconfidence is an inherent aspect of neural network models. Despite this, our CEMA demonstrates a robust capability to effectively filter out unreliable and low-informative samples. This effectiveness indicates that the impact of overconfidence on our CEMA is limited. Therefore, even in the presence of inherent overconfidence in neural networks, the approach adopted by CEMA to assess uncertainty and selectively upload test samples is validated.

## C MORE IMPLEMENTATION DETAILS

### C.1 MORE DETAILS ON DATASETS

**ImageNet-C**[1]. We evaluate our method and the counterparts on ImageNet-C (Hendrycks & Dietterich, 2019), which is a widely used benchmark dataset for out-of-distribution generalization. It is built based on the validation set of the original ImageNet by corrupting the images. Concretely, as shown in Figure 7, ImageNet-C includes 15 different corruption types, *i.e.*, Gaussian noise, shot noise, impulse noise, defocus blur, glass blue, motion blur, zoom blur, snow, frost, fog, brightness, contrast, elastic transformation, pixelation, and JPEG compression. Each corruption has five different severity levels (*i.e.*, from level 1 to level 5). Note that the larger severity level indicates a more severe distribution shift.

**ImageNet-R**[2]. We also evaluate our CEMA and compared methods on ImageNet-R (Hendrycks et al., 2021), which contains 30,000 images with various artistic renditions of 200 ImageNet classes, which are primarily collected from Flickr and filtered by Amazon MTurk annotators.

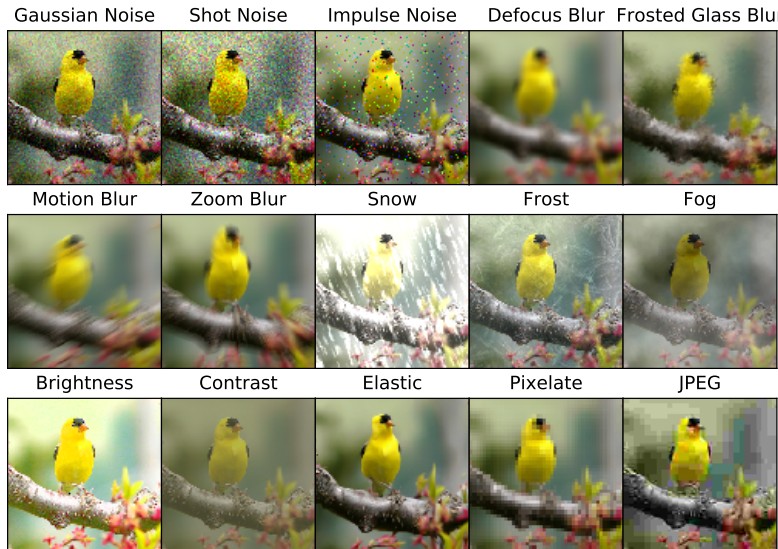

Figure 7: Visualizations of corrupted images with 15 corruption types in ImageNet-C benchmark, which are taken from the original paper of ImageNet-C (Hendrycks & Dietterich, 2019). Each corruption type has 5 severity levels, resulting in 75 distinct corruptions.

### C.2 MORE EXPERIMENTAL PROTOCOLS

**CEMA (Ours)**. In sample filtration strategy of edge devices, we set the entropy threshold $E_{\max}=0.4 \times \ln C$ as the initialized value following (Niu et al., 2022), where $C$ denotes the number of classes. Then the threshold $E_{\max}$ decreases based on Eqn. (2) with $\lambda=1.0$. We set another entropy threshold $E_{\min}=0.02 \times \ln C$ in the experiments. In addition to the removal of high/low-entropy test samples, we use the criteria in (Niu et al., 2022) to remove those samples with similar gradients (called *Non-redundant Sample Identification* in the original paper) for all the experiments. For the test-time adaptation of both the foundation model $f_{\theta(\cdot)}$ and the edge model $g_w(\cdot)$, we use an SGD optimizer with a learning rate of $0.00025$ and a momentum of $0.9$. We set the batch size to 32 while optimizing the foundation model.

For the adaptation of the edge model, we set the batch size to 128, in which 32 samples are newly uploaded and the remaining 96 samples are randomly sampled from the replay buffer. We set the hyper-parameters $\alpha$ and $\beta$ is set to 3 and 3, respectively. We set the replay buffer size to 10,000. Note

---

[1]https://github.com/hendrycks/robustness
[2]https://github.com/hendrycks/imagenet-r

that all the models used in our experiments are pretrained on ImageNet training set and available publicly. Specifically, ResNet18, ResNet101, MobileNetV2, MobileNetV3 and ShuffleNetV2 are from `torchvision`.[3] DeiT-tiny and DeiT-base are from `facebookresearch/deit`.[4] CLIP-ViT-B/32 is from `openai/CLIP`.[5] Since the edge model $g_w(\cdot)$ may infer with a small batch size (*e.g.*, the batch size is 1), we introduce Batch Renormalization (Ioffe, 2017) to replace the vanilla Batch Normalization in the edge model following (Zhao et al., 2023). In this case, $g_w(\cdot)$ is able to infer over a very small batch of test samples with moving average statistics instead of batch re-computing statistics.

**Compared methods.** We compare our methods with the following state-of-the-art TTA methods. BN Adaptation (Schneider et al., 2020) uses the weighted sum of training moving average BN statistics and batch re-computing statistics, in which both the batch size and prior strength are set to 256. ONDA (Mancini et al., 2018) adapts batch normalization statistics over a batch of test samples with an exponential moving average, in which the momentum is set to 0.9. Pseudo Label (PL) (Lee et al., 2013) adapts the model with the hard label generated by the model self. We train PL using an SGD optimizer with a learning rate of 0.001. Tent (Wang et al., 2021) updates the BN affine parameters via entropy minimization. The learning rate is set to 0.00025 and the batch size is set to 64. Based on Tent, ETA (Niu et al., 2022) removes test samples with high entropy and similar gradients. The entropy constant $E_0$ is set to $0.4 \times \ln C$, where $C$ is the number of task classes. The $\epsilon$ is set to 0.05. CoTTA (Wang et al., 2022) reduces the error accumulation by using weight-averaged and augmentation-averaged predictions. We use 32 augmentations and the augmentation threshold is set to 0.1. The learning rate is set to 0.01. The restoration probability is set to 0.01. LAME (Boudiaf et al., 2022) modifies the output probability of the classifier instead of the parameters of the model itself. We use the KNN kernel with 5 nearest neighbors.

---

[3] https://github.com/pytorch/vision
[4] https://github.com/facebookresearch/deit
[5] https://github.com/openai/CLIP

Table 6: Comparisons with state-of-the-art methods on ImageNet-C (severity levels 1, 2 and 4) regarding **Accuracy (%)**. We adopt Resnet101 as the foundation model and ResNet18 as the edge model. [†] denotes the TTA method that does not require any backpropagation and can be locally executed in edge devices. The **bold** number indicates the best result and the underlined number indicates the second-best result.

| | Noise | | | Blur | | | | Weather | | | | Digital | | | | |
| | Gauss. | Shot | Impul. | Defoc. | Glass | Motion | Zoom | Snow | Frost | Fog | Brit. | Contr. | Elastic | Pixel | JPEG | Avg. |
|---|---|---|---|---|---|---|---|---|---|---|---|---|---|---|---|---|
| **Severity Level=1** | | | | | | | | | | | | | | | | |
| ResNet18 (baseline) | 49.2 | 46.8 | 35.1 | 51.7 | 48.7 | 57.1 | 44.0 | 46.4 | 52.6 | 52.8 | 67.6 | 58.1 | 60.3 | 59.8 | 59.3 | 52.6 |
| • BN Adaptation[†] | 59.3 | 57.9 | 51.9 | 57.2 | 57.4 | 62.4 | 54.9 | 54.1 | 57.7 | 61.3 | 67.9 | 64.9 | 62.2 | 64.8 | 63.2 | 59.8 |
| • ONDA[†] | 58.7 | 58.0 | 51.3 | 55.2 | 56.1 | 62.3 | 55.0 | 53.6 | 57.2 | 61.9 | 68.3 | 64.6 | 62.3 | 65.1 | 63.0 | 59.5 |
| • LAME[†] | 48.8 | 46.3 | 34.1 | 51.4 | 48.2 | 56.8 | 43.6 | 45.9 | 52.3 | 52.5 | 67.3 | 57.8 | 60.1 | 59.5 | 58.9 | 52.2 |
| • PL | 60.5 | 60.3 | 55.9 | 58.8 | 60.2 | 63.6 | 58.0 | 57.5 | 59.0 | 63.1 | 67.7 | 65.4 | 62.6 | 65.3 | 63.5 | 61.4 |
| • Tent | 60.6 | 60.2 | 55.4 | 58.4 | 60.0 | 63.6 | 57.8 | 56.9 | 58.9 | 63.0 | 67.8 | 65.0 | 62.6 | 65.2 | 63.6 | 61.3 |
| • CoTTA | 59.1 | 58.7 | 52.6 | 56.7 | 57.4 | 62.8 | 56.0 | 54.5 | 57.6 | 62.2 | 67.8 | 64.6 | 62.5 | 65.1 | 63.1 | 60.0 |
| • ETA | **61.4** | **61.0** | 57.1 | 59.6 | **61.0** | **63.9** | 58.7 | 58.6 | 59.5 | **63.7** | 67.7 | 65.5 | **62.9** | 65.6 | 63.5 | **62.0** |
| • CEMA (Ours) | **61.7** | **61.5** | **58.2** | **59.8** | 60.8 | 63.5 | **58.8** | **59.2** | **59.6** | 63.6 | 67.1 | 65.2 | 62.5 | 65.3 | 63.4 | **62.0** |
| **Severity Level=2** | Gauss. | Shot | Impul. | Defoc. | Glass | Motion | Zoom | Snow | Frost | Fog | Brit. | Contr. | Elastic | Pixel | JPEG | Avg. |
| ResNet18 (baseline) | 37.9 | 33.8 | 25.8 | 44.2 | 36.4 | 45.2 | 34.2 | 23.4 | 35.0 | 46.0 | 65.6 | 51.2 | 39.1 | 59.7 | 55.8 | 42.2 |
| • BN Adaptation[†] | 52.5 | 49.6 | 45.5 | 50.4 | 48.7 | 55.7 | 48.6 | 39.4 | 45.1 | 58.3 | 66.8 | 62.7 | 46.2 | 63.8 | 60.2 | 52.9 |
| • ONDA[†] | 51.3 | 49.1 | 43.6 | 45.5 | 45.7 | 55.2 | 48.4 | 39.9 | 44.1 | 59.0 | 67.2 | 61.9 | 46.5 | 63.4 | 60.1 | 52.1 |
| • LAME[†] | 37.2 | 33.0 | 24.7 | 43.7 | 35.8 | 44.9 | 33.6 | 22.6 | 34.5 | 45.4 | 65.4 | 50.8 | 38.8 | 59.2 | 55.3 | 41.7 |
| • PL | 55.6 | 54.6 | 50.6 | 52.6 | 53.7 | 58.9 | 53.5 | 47.9 | 48.6 | 61.4 | 66.5 | 63.0 | 48.5 | 64.7 | 61.3 | 56.1 |
| • Tent | 55.1 | 54.0 | 49.7 | 52.0 | 52.7 | 58.7 | 52.9 | 46.3 | 48.3 | 61.0 | 66.6 | 63.0 | 48.4 | 64.4 | 61.2 | 55.6 |
| • CoTTA | 52.6 | 50.5 | 45.5 | 48.5 | 48.0 | 56.4 | 50.0 | 41.9 | 45.4 | 59.9 | 66.8 | 62.6 | 46.9 | 63.6 | 60.3 | 53.2 |
| • ETA | 57.0 | 56.1 | 52.6 | 54.4 | 55.2 | 59.8 | 54.8 | 50.0 | 50.2 | 62.2 | 66.6 | **64.0** | 49.1 | 64.9 | 61.4 | 57.2 |
| • CEMA (Ours) | **57.7** | **56.9** | **53.5** | **55.1** | **55.5** | 59.8 | 55.4 | **51.3** | **51.1** | 62.1 | 66.4 | 63.7 | **49.2** | 64.9 | 61.4 | **57.6** |
| **Severity Level=4** | Gauss. | Shot | Impul. | Defoc. | Glass | Motion | Zoom | Snow | Frost | Fog | Brit. | Contr. | Elastic | Pixel | JPEG | Avg. |
| ResNet18 (baseline) | 7.8 | 6.2 | 6.5 | 18.6 | 12.1 | 16.2 | 22.1 | 17.5 | 21.9 | 28.4 | 57.9 | 14.2 | 39.3 | 26.4 | 43.6 | 22.6 |
| • BN Adaptation[†] | 29.8 | 24.9 | 28.2 | 26.8 | 25.7 | 33.0 | 39.2 | 31.4 | 35.0 | 50.5 | 62.2 | 44.6 | 54.4 | 49.0 | 49.0 | 38.9 |
| • ONDA[†] | 26.9 | 23.6 | 24.8 | 19.0 | 21.4 | 30.7 | 38.7 | 31.2 | 33.8 | 51.4 | 62.3 | 37.4 | 54.0 | 49.5 | 48.7 | 36.9 |
| • LAME[†] | 6.9 | 5.2 | 5.1 | 18.3 | 11.4 | 15.8 | 21.4 | 16.8 | 21.3 | 27.3 | 57.6 | 13.4 | 38.6 | 25.6 | 43.1 | 21.8 |
| • PL | 38.3 | 36.1 | 36.6 | 30.6 | 33.7 | 40.7 | 45.9 | 40.3 | 38.2 | 56.0 | 62.5 | 44.0 | 59.0 | 54.9 | 53.9 | 44.7 |
| • Tent | 36.7 | 34.5 | 34.8 | 29.6 | 31.6 | 39.8 | 45.4 | 38.9 | 38.9 | 55.7 | 62.3 | 45.5 | 58.6 | 54.2 | 53.4 | 44.0 |
| • CoTTA | 29.2 | 25.6 | 27.1 | 21.2 | 23.3 | 34.7 | 41.4 | 32.8 | 35.4 | 53.2 | 62.5 | 43.8 | 54.9 | 51.6 | 50.1 | 39.1 |
| • ETA | **41.0** | 39.0 | 39.3 | 33.5 | 36.1 | 44.0 | 47.9 | 43.5 | 42.5 | 57.2 | 62.9 | 51.4 | 59.7 | 56.4 | 54.6 | 47.3 |
| • CEMA (Ours) | 38.5 | **40.4** | **41.2** | **35.4** | **38.4** | **45.1** | **48.6** | **44.9** | **43.1** | **58.0** | 62.9 | **52.6** | **60.0** | **56.9** | **55.7** | **48.0** |

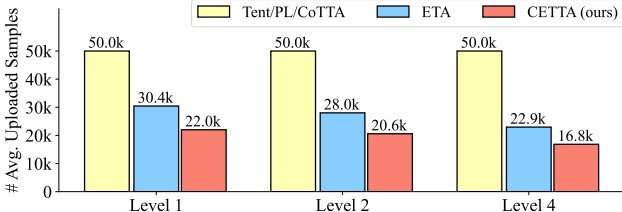

Figure 8: Comparisons with Tent, PL, CoTTA and ETA on CNN-based models in terms of the average number of uploaded test samples on ImageNet-C with the severity levels 1, 2 and 4.

## D    MORE EXPERIMENTAL RESULTS ON IMAGENET-C

### D.1    MORE COMPARISONS WITH CNN-BASED MODELS ON IMAGENET-C

In Table 6, we provide more results to compare our CEMA with state-of-the-art methods on ImageNet-C with the severity levels 1, 2 and 4. We adopt Resnet101 as the foundation (teacher) model and ResNet18 as the edge (student) model. Our CEMA achieves the highest average accuracy with severity levels 2 and 4. As for the corrupted images with level 1, our CEMA yields competitive performance with the best counterpart ETA. The possible reason is that the corruption of the images in level 1 is very slight. In this case, the foundation model may share a close performance with the edge model and be hard to transfer knowledge to it.

We also compare the communication overhead of our CEMA with Tent, PL, CoTTA and ETA in Figure 8. From the results, our CEMA needs to upload a smaller number of test samples to the cloud. For instance, our CEMA only requires 22.0k uploaded samples, which is fewer than Tent

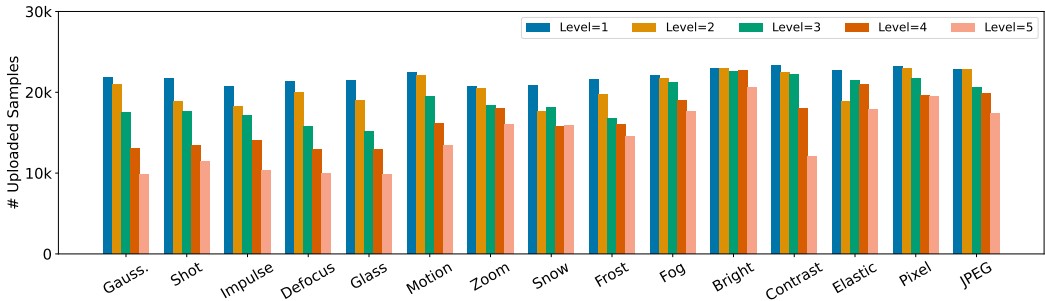

Figure 9: The number of uploaded test samples on ImageNet-C with different corruption types and severity levels.

(50.0k) and ETA (22.0k). The reason is that we exclude both high-entropy and low-entropy samples in the adaptation process. This improves efficiency in bandwidth-limited cloud-edge systems. The results demonstrate the superiority of our CEMA over the data transmission. In addition, we report the number of uploaded samples of our CEMA on ImageNet-C with different corruption types and severity levels in Figure 9.

## D.2    MORE COMPARISONS WITH TRANSFORMER-BASED MODELS ON IMAGENET-C

To verify the effectiveness of our CEMA on transformer-based models, we conduct more experiments on ImageNet-C using DeiT-base as the foundation model and Deit-tiny as the edge model. Since the baseline methods *BN Adaptation* and *ONDA* depends on batchnorm layers, we do not compare these two baselines on transformer-based models since these models have no batchnorm layers. Note that we only update the parameters in layernorm layers instead of batchnorm layers in transformer-based models.

From the results in Table 7, our CEMA outperforms the baselines Tent (44.5% *vs.* 21.8%), CoTTA (44.5% *vs.* 26.5%) and ETA (44.5% *vs.* 44.5%) greatly in severity level 5 on ImageNet-C. Since our CEMA 1) removes test samples that are harmful for the adaptation and 2) introduces a foundation model to transfer adapted knowledge to the edge model. Moreover, from Figure 10, our CEMA reduces more communication burden than the baseline methods, such as Tent, PL and ETA. For instance, our CEMA only requires uploading 13.9k test samples to the cloud, much lower than ETA (25.3k) and Tent (50.0k). The reason is that our CEMA devises an entropy-based sample filtration strategy, which excludes high/low entropy samples without helpful information. Note that Tent and PL achieve poor performance when the severity level is 5 since they adopt high-entropy samples with harmful information for adaptation. In sum, the above results on transformer-based models further verify the effectiveness of our CEMA.

## D.3    AVAILABILITY WITH CLIP FOUNDATION MODELS

We would like to highlight that it is common and practical that the foundation model possesses knowledge that covers the test samples inferred by the edge model. This can be achieved through several means: 1) simultaneously training a stronger foundation model and an edge model on the same training data; 2) using CLIP as the foundation model to circumvent the need to train one. Given the impressive ability for zero-shot classification of CLIP, it is likely to possess knowledge that covers test samples across extensive scenarios. For those cases that even CLIP cannot handle, the complexity exceeds the scope of our current research, and we would defer it to future studies.

In this section, we conduct more experiments on ImageNet-R to demonstrate that our CEMA works well with CLIP as the foundation model. Note that the CLIP model is pretrained based on its private data and does not access the training data of the edge model (ResNet18). In Table 8, we adopt CLIP-ViT-B/32 as the foundation model and ResNet18 as the edge model to perform cloud-edge adaptation. From the results, our CEMA consistently outperforms Tent and ETA. The results demonstrate that our CEMA integrates effectively with the CLIP foundation model in practice. We

Table 7: Comparisons with state-of-the-art methods on ImageNet-C regarding **Accuracy (%)**. We adopt DeiT-base as the foundation model and DeiT-tiny as the edge model. [†] denotes the TTA method that does not require any backpropagation and can be locally executed in edge devices. The **bold** number indicates the best result and the underlined number indicates the second-best result.

| | Noise | | | Blur | | | | Weather | | | | Digital | | | | |
|---|---|---|---|---|---|---|---|---|---|---|---|---|---|---|---|---|
| **Severity Level=1** | Gauss. | Shot | Impul. | Defoc. | Glass | Motion | Zoom | Snow | Frost | Fog | Brit. | Contr. | Elastic | Pixel | JPEG | Avg. |
| DeiT-tiny (baseline) | 63.0 | 62.8 | 60.7 | 56.5 | 55.8 | 63.4 | 46.1 | 57.4 | 62.4 | 59.3 | 69.6 | 65.0 | 64.0 | 61.2 | 62.4 | 60.6 |
| • LAME[†] | 62.8 | 62.5 | 60.5 | 56.0 | 55.2 | 63.0 | 45.4 | 57.0 | 62.0 | 59.0 | 69.2 | 64.7 | 63.6 | 60.8 | 62.1 | 60.3 |
| • PL | 64.4 | 64.3 | 62.5 | 61.7 | 62.0 | 65.8 | 54.6 | 60.3 | 63.7 | 64.0 | **70.4** | 67.3 | 66.0 | 65.3 | 64.2 | 63.8 |
| • Tent | 64.5 | 64.3 | 62.5 | 62.1 | 63.0 | 66.0 | 56.9 | 61.0 | 63.9 | 64.7 | **70.4** | 67.7 | 66.2 | 66.1 | 65.0 | 64.3 |
| • CoTTA | 63.5 | 63.2 | 61.3 | 57.2 | 56.7 | 63.9 | 46.9 | 58.3 | 63.0 | 60.4 | 69.8 | 65.9 | 64.4 | 62.1 | 63.1 | 61.3 |
| • ETA | 64.8 | 64.6 | 62.9 | 62.4 | 63.9 | 66.2 | 59.2 | 61.8 | **64.2** | **65.5** | 70.3 | **68.0** | **66.3** | 67.4 | 65.8 | 64.9 |
| • CEMA (Ours) | **65.3** | **64.9** | **63.2** | **63.0** | **64.4** | **66.5** | **60.1** | **62.5** | 64.1 | **65.5** | 70.2 | 67.8 | 66.2 | **68.0** | **66.2** | **65.2** |
| **Severity Level=2** | Gauss. | Shot | Impul. | Defoc. | Glass | Motion | Zoom | Snow | Frost | Fog | Brit. | Contr. | Elastic | Pixel | JPEG | Avg. |
| DeiT-tiny (baseline) | 58.4 | 57.0 | 54.5 | 50.5 | 45.0 | 56.0 | 37.4 | 41.3 | 52.5 | 54.1 | 68.4 | 63.2 | 43.6 | 56.2 | 59.0 | 53.1 |
| • LAME[†] | 58.2 | 56.8 | 54.2 | 50.0 | 44.2 | 55.6 | 36.6 | 40.8 | 52.1 | 53.4 | 68.1 | 63.0 | 43.2 | 55.8 | 58.7 | 52.7 |
| • PL | 60.2 | 59.6 | 57.5 | 56.6 | 54.8 | 61.1 | 47.4 | 48.0 | 54.5 | 62.2 | 69.5 | 66.0 | 49.0 | 63.6 | 61.4 | 58.1 |
| • Tent | 60.5 | 59.7 | 57.6 | 57.4 | 56.7 | 61.8 | 50.7 | 50.0 | 55.7 | 63.3 | 69.5 | 66.3 | 50.9 | 64.6 | 62.4 | 59.1 |
| • CoTTA | 59.0 | 57.7 | 55.2 | 51.1 | 45.8 | 57.0 | 38.1 | 42.3 | 53.4 | 55.6 | 68.7 | 64.2 | 44.3 | 57.3 | 59.7 | 54.0 |
| • ETA | 60.9 | 60.2 | 58.2 | 57.9 | 58.7 | 62.3 | 54.7 | 52.3 | 56.8 | **64.1** | **69.7** | **67.0** | 53.0 | 66.4 | 63.6 | 60.4 |
| • CEMA (Ours) | **61.4** | **60.6** | **58.9** | **58.6** | **59.2** | **62.7** | **55.8** | **54.1** | **57.4** | 63.9 | 69.4 | 66.6 | **53.6** | **67.2** | **64.2** | **60.9** |
| **Severity Level=3** | Gauss. | Shot | Impul. | Defoc. | Glass | Motion | Zoom | Snow | Frost | Fog | Brit. | Contr. | Elastic | Pixel | JPEG | Avg. |
| DeiT-tiny (baseline) | 49.1 | 48.0 | 48.6 | 38.1 | 20.5 | 43.8 | 31.6 | 44.9 | 44.2 | 47.0 | 66.7 | 60.6 | 55.5 | 47.6 | 56.8 | 46.9 |
| • LAME[†] | 48.9 | 47.7 | 48.3 | 37.5 | 19.2 | 43.5 | 30.8 | 44.3 | 43.8 | 46.2 | 66.4 | 60.3 | 55.1 | 47.0 | 56.4 | 46.3 |
| • PL | 52.7 | 52.8 | 53.1 | 46.1 | 35.6 | 53.3 | 42.4 | 49.8 | 46.9 | 58.4 | 67.9 | 63.7 | 62.3 | 58.4 | 59.6 | 53.5 |
| • Tent | 53.1 | 53.1 | 53.4 | 47.9 | 41.0 | 54.7 | 46.3 | 51.5 | 48.2 | 60.0 | 68.1 | 64.1 | 63.8 | 60.1 | 60.7 | 55.1 |
| • CoTTA | 49.8 | 48.8 | 49.4 | 39.0 | 20.9 | 45.1 | 32.1 | 46.0 | 45.4 | 49.0 | 67.0 | 61.6 | 56.5 | 49.0 | 57.5 | 47.8 |
| • ETA | 54.1 | 54.2 | 54.2 | 49.4 | 47.0 | 56.1 | 51.7 | 53.7 | 51.0 | **61.5** | 68.1 | **64.6** | 64.7 | 62.4 | 62.0 | 57.0 |
| • CEMA (Ours) | **55.0** | **55.1** | **55.1** | **50.5** | **48.5** | **57.1** | **52.9** | **55.4** | **51.8** | 60.2 | **68.4** | 64.3 | **65.5** | **63.4** | **63.0** | **57.7** |
| **Severity Level=4** | Gauss. | Shot | Impul. | Defoc. | Glass | Motion | Zoom | Snow | Frost | Fog | Brit. | Contr. | Elastic | Pixel | JPEG | Avg. |
| DeiT-tiny (baseline) | 35.9 | 30.4 | 33.4 | 27.6 | 16.1 | 30.2 | 26.0 | 36.5 | 43.2 | 41.6 | 63.8 | 49.0 | 44.7 | 25.8 | 49.2 | 36.9 |
| • LAME[†] | 35.6 | 30.1 | 33.0 | 27.0 | 14.7 | 29.7 | 24.9 | 35.9 | 42.8 | 39.0 | 63.6 | 48.6 | 44.0 | 25.2 | 48.8 | 36.2 |
| • PL | 42.2 | 39.7 | 42.0 | 34.3 | 4.5 | 43.5 | 10.7 | 42.1 | 42.3 | 2.7 | 65.8 | 55.9 | 55.3 | 49.2 | 53.5 | 38.9 |
| • Tent | 43.1 | 41.0 | 42.8 | 38.1 | 4.5 | 45.4 | 39.1 | 45.1 | 47.1 | 4.3 | 66.0 | 56.7 | 58.3 | 52.4 | 55.0 | 42.6 |
| • CoTTA | 36.6 | 31.4 | 34.1 | 28.2 | 16.3 | 31.5 | 26.4 | 37.5 | 44.3 | 44.0 | 64.5 | 50.8 | 45.6 | 26.7 | 50.0 | 37.9 |
| • ETA | 44.9 | 43.0 | 44.8 | 41.4 | 42.2 | 48.3 | 47.7 | 48.2 | 50.0 | **59.4** | 66.3 | **57.6** | 61.2 | 55.8 | 57.2 | 51.2 |
| • CEMA (Ours) | **46.7** | **45.1** | **46.2** | **42.6** | **43.6** | **50.1** | **48.4** | **50.4** | **50.8** | 56.4 | **66.5** | 56.3 | **62.0** | **57.7** | **58.6** | **52.1** |
| **Severity Level=5** | Gauss. | Shot | Impul. | Defoc. | Glass | Motion | Zoom | Snow | Frost | Fog | Brit. | Contr. | Elastic | Pixel | JPEG | Avg. |
| DeiT-tiny (baseline) | 17.0 | 18.2 | 17.4 | 19.2 | 12.6 | 22.9 | 20.9 | 32.6 | 37.6 | 32.9 | 59.6 | 23.9 | 23.5 | 10.3 | 38.5 | 25.8 |
| • LAME[†] | 16.5 | 17.9 | 17.0 | 18.6 | 11.4 | 22.4 | 19.9 | 31.4 | 37.1 | 29.6 | 59.3 | 23.4 | 21.3 | 10.1 | 38.1 | 24.9 |
| • PL | 1.0 | 2.3 | 1.1 | 17.8 | 2.4 | 35.9 | 3.6 | 9.4 | 15.2 | 0.8 | 62.8 | 38.6 | 3.9 | 35.9 | 46.2 | 18.5 |
| • Tent | 4.1 | 13.3 | 13.6 | 27.1 | 1.6 | 38.7 | 3.4 | 11.7 | 14.6 | 0.8 | 63.2 | **41.3** | 2.4 | 44.1 | 47.8 | 21.8 |
| • CoTTA | 17.6 | 18.8 | 18.1 | 19.7 | 12.7 | 23.9 | 21.0 | 33.7 | 38.7 | 34.8 | 60.4 | 24.4 | 24.1 | 10.6 | 39.3 | 26.5 |
| • ETA | 32.1 | 33.7 | 33.4 | 33.8 | **35.0** | 42.9 | **43.4** | 45.9 | 46.0 | 53.2 | 63.9 | 33.9 | **50.2** | **50.6** | 51.0 | 43.1 |
| • CEMA (Ours) | **34.4** | **36.7** | **36.2** | **35.8** | 34.4 | **44.8** | 43.0 | **48.0** | **46.8** | **54.6** | **64.0** | 37.0 | 49.9 | 50.1 | **52.8** | **44.5** |

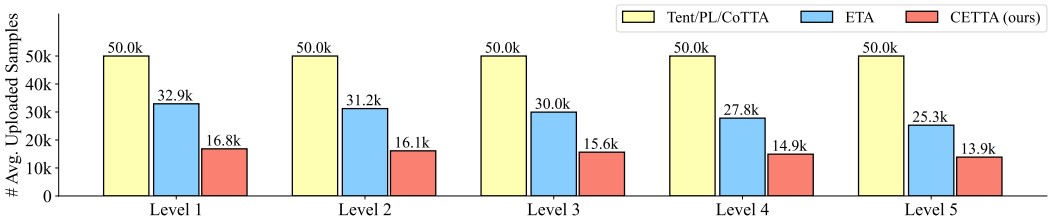

Figure 10: Comparisons with Tent, PL, CoTTA and ETA on Transformer-based models in terms of the average number of uploaded test samples on ImageNet-C with the severity levels 1 - 5.

also verify that the edge model is able to be guided by different simultaneously training foundation models in Table 13.

Table 8: Comparisons with Tent and ETA on ImageNet-R. We adopt CLIP-ViT-B/32 as the foundation model and ResNet18 as the edge model.

| Model | Acc. (%) | #Uploaded samples |
|---|---|---|
| ResNet18 (baseline) | 20.4 | – |
| • Tent (Wang et al., 2021) | 23.6 | 30,000 (100%) |
| • ETA (Niu et al., 2022) | 26.2 | 7,457 (25%) |
| • CEMA (Ours) | **26.9** | **6,521 (22%)** |

# E    MORE ABLATION RESULTS

## E.1    EFFECT OF $E_{\text{MAX}}$ IN EQN. (1)

Our CEMA employs hyperparameters $E_{\text{max}}$ and $E_{\text{min}}$ and in Eqns. (1) and (3), respectively. These two hyperparameters denote the thresholds while excluding high/low entropy test samples. In practice, we determine these hyperparameters through a systematic approach guided by the following principles: 1) We set these hyperparameters to be linked to the number of classes instead of grounded in absolute terms. Specifically, we prescribe its value as $\gamma \times lnC$, where $0 < \gamma < 1$ and $C$ denotes the number of classes. Consequently, this hyperparameter exhibits a robust insensitivity to fluctuations in the number of classes. 2) Next we select the appropriate values of these hyperparameters by examining the impact of hyperparameters on the ImageNet-C (Gaussian noise, severity level 3) dataset. Based on the results, we select relatively well-performing hyperparameters and keep them as constants by default across different datasets, including ImageNet-R and various corruption types and severity levels within ImageNet-C.

In our entropy-based sample filtration, we adopt an entropy threshold $E_{\text{max}}$ to remove high-entropy samples. Since the high-entropy samples may have a negative impact on adaptation performance. In Table 9, we report the adaptation performance when we upload less/more high entropy samples by adjusting $E_{\text{max}}$. When $E_{\text{max}}$ is small, our CEMA removes too many samples during adaptation and thus it is hard to learn helpful knowledge from the remaining samples. When $E_{\text{max}}$ is too large, some high-entropy samples would participate in the adaptation and contribute harmful gradients, resulting in performance degradation. From the results, our CEMA achieves the highest adaptation performance when $E_{\text{max}}$=0.4. Thus we set $E_{\text{max}}$ to 0.4 in our experiments.

Table 9:    Effect of the entropy threshold $E_{\text{max}}$ of our CEMA on ImageNet-C (Gaussian noise, severity level=3).

| $E_{\text{max}}$ | 0.2 | 0.3 | 0.4 | 0.5 | 0.6 |
|---|---|---|---|---|---|
| Accuracy (%) | 50.2 | 50.8 | 51.1 | 50.5 | 46.4 |
| #Uploading samples | 10,560 | 14,276 | 17,479 | 20,067 | 21,931 |

## E.2    EFFECT OF $\alpha$ AND $\beta$ IN EQN. (6)

We conduct ablation studies to investigate the effect of different hyper-parameters $\alpha$ and $\beta$ in Eqn. (6) in Table 10 and 11, respectively. Here, $\alpha$ and $\beta$ both are selected from $\{1, 2, 3, 4, 5, 6\}$. From the results, when $\alpha$=3 our CEMA achieves the best performance (51.1%). As for the hyper-parameter $\beta$, we obtain the highest accuracy (51.1%) when $\beta$=3. Thus, we set $\alpha$=3 and $\beta$=3 in all the experiments. In other scenarios on various datasets, we keep them the same as those on Gaussian noise, including ImageNet-R and ImageNet-C (encompassing 15 types of corruption and 5 severity levels, 75 different scenarios in total). Extensive experiments show that our CEMA works well with these hyperparameters.

Table 10:    Effect of hyper-parameter $\alpha$ in Eqn. (6) on ImageNet-C (Gaussian noise, severity level 3).

| $\alpha$ | 1 | 2 | 3 | 4 | 5 | 6 |
|---|---|---|---|---|---|---|
| Accuracy (%) | 50.5 | 51.0 | 51.1 | 51.0 | 50.6 | 50.3 |

Table 11:    Effect of hyper-parameter $\beta$ in Eqn. (6) on ImageNet-C (Gaussian noise, severity level 3).

| $\beta$ | 1 | 2 | 3 | 4 | 5 | 6 |
|---|---|---|---|---|---|---|
| Accuracy (%) | 50.7 | 51.0 | 51.1 | 50.7 | 50.4 | 50.0 |

## E.3    EFFECT OF KNOWLEDGE DISTILLATION LOSS

While updating the edge model via Eqn. (6), entropy minimization is not equal to optimizing with the pseudo label from its predictions. Since the entropy minimization and pseudo label approaches serve different functions to improve the performance of the edge model. The pseudo labels provide a direct label for the corresponding sample to align the decision boundaries. On the other hand, entropy

minimization encourages the model to make more confident predictions by penalizing uncertainty in its output distribution. Furthermore, it is important to clarify that the entropy minimization approach operates on the average entropy across the entire batch of samples, rather than focusing on the confidence of individual samples. This methodology aims to optimize the overall confidence of the model, guiding it towards more assured predictions on a collective basis, rather than encouraging each sample to converge towards its highest predictive probability. This distinction underscores the complementary nature of our CEMA, utilizing both cross-entropy loss for decision boundary alignment to the foundation model and entropy loss for enhancing the model's collective confidence.

In Table 12, we report the performance of our CEMA with/without $\mathcal{L}_{\text{KL}}$ as well as $\mathcal{L}_{\text{CE}}$ in Eqn. (6). The baseline without $\mathcal{L}_{\text{KL}}$ and$\mathcal{L}_{\text{CE}}$ employs only $\mathcal{L}_{\text{ENT}}$. From the results, with the KL divergence $\mathcal{L}_{\text{KL}}$, our CEMA achieves better accuracy than the baseline ($\mathcal{L}_{\text{ENT}}$) (50.5% *vs.* 50.0%). Since the foundation model transfers its knowledge on out-of-distribution samples to the edge model via $\mathcal{L}_{\text{KL}}$. In addition, our CEMA further improves performance with $\mathcal{L}_{\text{CE}}$. The results show that the combination of both losses outperforms the adoption of either one in isolation (51.1% *vs.* 50.5%). It highlights the necessity and significance of the CE loss.

Table 12: Effect of components in the loss for updating the edge model on ImageNet-C (Gaussian noise) with ResNet18 as the edge model.

| $\mathcal{L}_{\text{ENT}}$ | $\mathcal{L}_{\text{CE}}$ | $\mathcal{L}_{\text{KL}}$ | Accuracy (%) | #Uploads |
|:---:|:---:|:---:|:---:|:---:|
| ✗ | ✓ | ✓ | 50.2 | 18,497 |
| ✓ | ✗ | ✗ | 50.0 | 17,263 |
| ✓ | ✗ | ✓ | 50.5 | 17,379 |
| ✓ | ✓ | ✗ | 50.7 | 17,315 |
| ✓ | ✓ | ✓ | **51.5** | 17,479 |

### E.4 POTENTIAL OF STRONG FOUNDATION MODELS.

In the cloud, we can exploit stronger foundation models for adaptation when we have more computational budgets. In this sense, our CEMA has great potential in real-world applications. In Table 13, we report the results on ImageNet-C (Gaussian noise, severity level 3) with different foundation models, namely, ResNet101, ResNet152 and ConvNeXt-T (Liu et al., 2022). Note that ResNet152 and ConvNeXt-T are stronger foundation models, which outperform ResNet101 on ImageNet. From the results, as the foundation model becomes stronger, the edge model yields higher accuracy. The results show the potential of CEMA with stronger foundation models in real-world applications.

Table 13: Effect of different foundation models.

| Foundation Model | Level 3 | | Level 5 | |
|:---|:---:|:---:|:---:|:---:|
| | Acc. (%) | #Upload | Acc. (%) | #Upload |
| ResNet101 | 51.1 | 17,479 | 29.8 | 9,889 |
| ResNet152 | 51.5 | 17,461 | 30.3 | 10,484 |
| ConvNeXt-T | 52.0 | 17,936 | 31.2 | 10,236 |

### E.5 APPLICABILITY TO DIFFERENT EDGE MODELS.

In Table 14, we report the results on ImageNet-C (Gaussian noise, severity level 5) with different light-weight models that can be deployed on the resources-limited edge, including MobileNetV2, MobileNetV3 and ShuffleNetV2. On all these edge models, our CEMA outperforms ETA in terms of both the adaptation accuracy and the communication overhead. For example, our CEMA achieves higher accuracy (37.1% *vs.* 31.1%) than ETA with ShuffleNetV2 as the edge model with a much fewer number of uploaded samples (13,975 *vs.* 24,558). The results show that our CEMA is applicable to various lightweight edge models.

Table 14: Comparisons of different edge models on ImageNet-C (Gaussian noise, severity level 5) with ResNet101 as the foundation model. We adopt two popular lightweight edge models, *i.e.*, MobileNetV2, MobileNetV3 and ShuffleNetV2, to verify our proposed method CEMA.

| Model | Acc. (%) | #Upload |
|---|---|---|
| MobileNetV2 (baseline) | 19.4 | – |
| • ETA (Niu et al., 2022) | 45.6 | 25,133 (50%) |
| • CEMA (Ours) | **47.3** | **16,117 (32%)** |
| MobileNetV3 (baseline) | 25.3 | – |
| • ETA (Niu et al., 2022) | 40.1 | 19,094 (38%) |
| • CEMA (Ours) | **41.2** | **15,414 (31%)** |
| ShuffleNetV2 (baseline) | 11.2 | – |
| • ETA (Niu et al., 2022) | 31.3 | 24,668 (49%) |
| • CEMA (Ours) | **37.1** | **13,975 (28%)** |

### E.6 EFFECTIVENESS OF THE ENTROPY-BASED CRITERIA

To verify the effectiveness of the proposed entropy-based sample filtration strategy, we compare our CEMA with two variants on ImageNet-C (Gaussian noise, severity level 5), namely *uploading all test samples* and *randomly uploading an equal number of test samples*. The variant *randomly uploading an equal number of test samples* denotes randomly uploading only partial test samples to the cloud, in which the number of uploading samples is the same as our CEMA. From Table 15, our CEMA outperforms *randomly uploading an equal number of test samples* (29.8% *vs.* 28.0%). In addition, though the variant *uploading all test samples* achieves higher performance since it provides sufficient test samples for distillation. It requires many more test samples to be uploaded from the edge to the cloud (50,000 *vs.* 9,889). The results demonstrate the effectiveness of our devised entropy-based filtering criteria.

Table 15: Effectiveness of the proposed entropy-based criteria on ImageNet-C (Gaussian noise, severity level 5) with ResNet18 as the edge model.

| Strategy | Accuracy (%) | #Uploading Samples |
|---|---|---|
| Uploading all test samples | 30.6 | 50,000 |
| Randomly uploading an equal number of test samples | 28.0 | 10,153 |
| Uploading with entropy-based criteria (our CEMA) | 29.8 | 9,889 |

### E.7 EFFECT OF THE REPLAY BUFFER SIZE

To investigate the effect of the size of the replay buffer, we conduct an ablation with the different sizes selected from $\{0, 1000, 2000, 3000, 5000, 10000, \infty\}$. Note that the size $\infty$ denotes the buffer is unlimited. From Table 16, our CEMA achieves better accuracy on ImageNet-C when we increase the replay buffer size. We obtain the best performance at 51.1% when the buffer size is 10,000 (10 samples for each class on average). Employing a replay buffer of unlimited size does not yield any improvement in adaptation accuracy, which remains at 51.1%. However, this leads to a significant increase in storage usage, escalating from 5.6 GB to 9.8 GB. The reason is that we can sample more diverse samples for the larger replay buffer. This may provide more distribution information for adaptation. Note that a replay buffer with a size of 10,000 with image resolution 224×224 requires around 5.6 GB memory. This can be affordable for the typical cloud GPU servers. Thus we set the buffer size to 10,000 in all the experiments.

Table 16: Effect of the size of the replay buffer $\mathcal{B}$ on ImageNet-C (Gaussian noise, severity level 3) with ResNet18 as the edge model.

| Size | 0 | 1000 | 2000 | 3000 | 5000 | 10000 | $\infty$ |
|---|---|---|---|---|---|---|---|
| Accuracy (%) | 47.7 | 50.0 | 50.3 | 50.5 | 50.9 | 51.1 | 51.1 |
| Storage (GB) | 0 | 0.6 | 1.1 | 1.7 | 2.8 | 5.6 | 9.8 |

### E.8 EFFECT OF DIFFERENT UPDATING WAYS

To investigate the effect of different ways to update parameters, we conduct more experiments on ImageNet-C (Gaussian noise, severity level 3). We consider two ways to update models: 1) updating all the parameters and 2) only updating parameters in batchnorm (BN) layers. From Table 17, we see only updating BN on the foundation model and the edge model achieves the best adaptation performance and the lowest communication overhead (especially on distributed parameter size). Updating all the layers may disrupt the previously learned knowledge and lead to inferior adaptation performance. Furthermore, it would require the distribution of more parameters and result in significant communication overhead. Thus, we choose to only update BN layers for both the foundation model and the edge model.

Table 17: Ablations the effects of different ways to update parameters on ImageNet-C (Gaussian noise, severity level 3). $f_\theta$ and $f_w$ denote the foundation model and the edge model, respectively. "Distributed Param. Size" denotes how many model parameters should be transferred between the cloud and the edge.

| Only updating BN for $f_\theta$ | Only updating BN for $f_w$ | #Upload | Distributed Param. Size (MB) | Acc. (%) |
|---|---|---|---|---|
| ✗ | ✗ | 14,221 | 11.68 | 31.9 |
| ✗ | ✓ | 16,871 | 0.0096 | 46.4 |
| ✓ | ✗ | 17,608 | 11.68 | 43.6 |
| ✓ | ✓ | 17,479 | 0.0096 | 51.1 |

### E.9 EFFECT OF UPDATING INTERVAL IN EDGE

In scenarios where communication and computation resources are limited, the edge devices may only download and update the edge model once while the cloud performs every $K$ times adaptation ($K>1$). To investigate the effect of $K$, we perform more experiments on ImageNet-C (Gaussian noise, severity level 3) with different $K$ from 1 to 5. From Table 18, the adaptation performance slightly drops when $K$ grows. For example, when $K$ increases from 1 to 3, the adaptation accuracy only drops from 51.1% to 50.8%. Even when $K$ becomes 5, the adaptation accuracy is still 50.5%. These demonstrate the effectiveness of our CEMA in scenarios with limited bandwidth.

Table 18: Performance comparisons on ImageNet-C (Gaussian noise, severity level 3) with different updating intervals $K$ on the edge side.

| $K$ | 1 | 2 | 3 | 4 | 5 |
|---|---|---|---|---|---|
| Accuracy (%) | 51.1 | 51.0 | 50.8 | 50.7 | 50.5 |
| #Uploading samples | 17,479 | 17,492 | 17,333 | 17,396 | 17,402 |

### E.10 COMPARISONS WITH MORE SAMPLE IDENTIFICATION STRATEGIES

To further demonstrate the effectiveness of our sample identification strategy, we add more experiments on ImageNet-C to compare our CEMA with more strategies, namely SENTRY (Prabhu et al., 2021) and BALD (Houlsby et al., 2011). SENTRY measures the sample information via the prediction consistency regarding different data augmentations. Besides, BALD achieves this by calculating the entropy differences between the current sample and previous samples.

From Table 19, SENTRY achieves much worse accuracy than our CEMA (45.6% *vs.* 51.1%). The results show that the prediction consistency regarding different data augmentations is hard to identify the helpful samples in entropy minimization. Besides, BALD yields an adaptation accuracy of 50.7%, which is still worse than our CEMA. The results demonstrate the effectiveness of our CEMA in filtering out low-informative samples.

Table 19: Comparisons of different strategies to identify unreliable and low-informative samples on ImageNet-C (Gaussian noise) with ResNet18 as the edge model.

| Strategy | Accuracy (%) | #Uploads |
|---|---|---|
| SENTRY (Prabhu et al., 2021) | 45.6 | 17,299 |
| BALD (Houlsby et al., 2011) | 50.7 | 17,319 |
| CEMA (Ours) | **51.1** | 17,479 |

### E.11 MORE COMPARISONS ON OBJECT DETECTION

We conduct more experiments by applying our CEMA on corrupted COCO 2017 (Lin et al., 2014) with YOLOv5 (Redmon et al., 2016) model. We generate this corrupted version (Gaussian noise, severity level 3) of COCO 2017 dataset following ImageNet-C (Hendrycks & Dietterich, 2019). We use YOLOv5-nano as the edge model and YOLOv5-large as the foundation model. We set the learning rate to 0.0001 and the other hyperparameters are the same as those in the image classification task. From Table 20, our CEMA outperforms the baseline (18.3 *vs.* 11.6 mAP) and ETA (18.3 *vs.* 15.0 mAP). Moreover, this improved performance was achieved with a smaller number of samples uploaded for adaptation. The results demonstrate the applicability and effectiveness of our CEMA on the object detection task.

Table 20: Comparisons on corrupted COCO 2017 (Gaussian noise, severity level 3).

| Model | mAP | #Upload |
|---|---|---|
| YOLOv5-nano (baseline) | 11.6 | – |
| • ETA (Niu et al., 2022) | 15.0 | 3,875 (78%) |
| • CEMA (Ours) | **18.3** | **3,261 (65%)** |

