# OpenReview forum: "Towards Robust and Efficient Cloud-Edge Elastic Model Adaptation via Selective Entropy Distillation"
_ICLR.cc/2024/Conference — ICLR 2024 poster_

### Official Review · Reviewer_Q1Z2 · 2023-10-30

**Soundness:** 3 good
**Presentation:** 3 good
**Contribution:** 3 good
**Rating:** 5
**Confidence:** 3

**Summary:**

This paper focuses on adapting deep learning models for edge devices with limited resources in dynamic environments. Traditional approaches involve deploying fixed models, which can result in reduced performance as scenarios change. This paper devises the Cloud-Edge Model Adaptation (CEMA) paradigm, in which the edge models only need to perform forward propagation and the edge models can be adapted online, by performing a data filtering strategy to allow high-quality data to be uploaded to the cloud and a replay-based entropy distillation.

**Strengths:**

1. This paper is well-written, easy to follow.
2. Manage what data to train is a sound approach to reduce training efficiency.
3. Experiments are comprehensive.

**Weaknesses:**

From algorithmic perspective (low-informative data identification), the novelty is limited. There should be existing papers studied how to filter out low-informative data. These can be added into the paper related works and experiments to compare.

**Questions:**

1. In equation 6, is fθ(x) and the pseudo labels yˆ  the same thing or different?
2. To use KL divergence loss and CE loss together is interesting. I wonder how the hyperparameter alpha and beta change in different dynamic scenarios.

---

> ### Author Response · Authors · 2023-11-20
> **Responses to Reviewer Q1Z2 [1/2]**
>
> We greatly appreciate the time and effort you have dedicated to reviewing our manuscript. Your insights and comments are invaluable in enhancing the quality of our work. We welcome any further questions or concerns you may have, as we are eager to address them and improve our manuscript accordingly.
>
> ---
>
> >Q1. Significance and Novelty of our CEMA.
>
> A1. In this work, we address a very **practical and challenging** problem, **cloud-edge model deployment under distribution-shifted scenarios**. Recently, the use of remote models in the cloud has gained popularity in model deployment, as it has been shown to improve performance and reduce computational resource costs locally. More critically, we may have to deal with distribution shifts on test data, which are commonly occurred in dynamic and complex real-world scenarios. Motivated by these, we seek to investigate how to efficiently adapt models with distribution-shifted and unlabeled test data in cloud-edge systems.
>
>
> We would like to highlight our **significance and novelty** in the following two folds:
>
> 1. **New Cloud-edge Model Deployment Setting under Distribution-shifted Scenarios**.
>    - We study cloud-edge model deployment in a **realistic setting**, in which test data are distribution-shifted from the training data. This setting is **quite common in real-world applications** and very challenging. Since it is hard to perform model adaptation on edge devices (such as surveillance cameras in industrial parks) due to limited resources. In this case, it is necessary to leverage foundation models in the cloud to perform adaptation in a collaborative way.
>
> 2. **New Collaborative Cloud-edge Model Adaptation Paradigm for Robustness Enhancement**
>
>    - We establish a novel cloud-edge model adaptation (CEMA) paradigm that collaboratively adapts models by leveraging **both the cloud and the edge**. Our framework is **highly efficient in edges** since it only conducts vanilla forward propagation without training.
>    - We devise two entropy-based criteria to reduce data transmission overhead by excluding unreliable and low-informative samples while uploading samples. Furthermore, we propose a replay-based distillation method to exploit the massive resources in the cloud. Experimental results demonstrate our CEMA not only **reduces 60% data traffic** but also **outperforms SOTA adaptation methods** on the ImageNet-C benchmark dataset.

---

> ### Author Response · Authors · 2023-11-20
> **Responses to Reviewer Q1Z2 [2/2]**
>
> >Q2. From algorithmic perspective (low-informative data identification), the novelty is limited. There should be existing papers studied how to filter out low-informative data. These can be added into the paper related works and experiments to compare.
>
> A2. Thank you for your insightful suggestions. We clarify our differences compared with existing low-informative sample identification methods in the following:
>
> **Differences from Existing Low-informative Sample Identification Approaches**: In our CEMA, we adopt the entropy to measure the sample's information, as it is easy to use and efficient. Unlike existing methods used in active learning and domain adaptation that may rely on a whole dataset [A] or less-efficient multiple forward propagations [B], the entropy can be calculated over a single sample and only involves one-time forward propagation. For example, Sener et al. [A] propose a core-set approach to select samples that can mostly represent the whole dataset. SENTRY [B] measures the sample information via the prediction consistency regarding different data augmentations. We have added more discussions regarding this in the related work section.
>
> Compared with ETA[C] which also exploits entropy for sample identification, our sample selection strategy is different in the two aspects: 1) ETA adopts a static threshold to select unreliable samples, while we reveal that the suitable threshold may change continuously along with the online adaptation process and propose a dynamic unreliable sample identification strategy; 2) we further introduce a new low-informative sample selection strategy to identify samples that produce negligible gradients for model updating. The empirical comparisons with ETA have been presented in Tables 1-3 in our manuscript.
>
> We would like to point out that identifying low-informative samples is only a mere aspect of our paper, as we re-clarified our main contributions and novelty at the beginning of the responses.
>
> **Empirical Comparisons**: To further demonstrate the effectiveness of our sample identification strategy, we add more experiments on ImageNet-C to compare our CEMA with SENTRY [B] and ETA [C]. From Table A, SENTRY achieves worse accuracy than our CEMA (45.6% vs. 51.1%). The results show the prediction consistency regarding different data augmentations is hard to identify the helpful samples in entropy minimization. Besides, ETA yields an adaptation accuracy of 50.1%, which is still worse than our CEMA. Since ETA only filters out a part of unreliable samples by employing a fixed threshold. The results demonstrate the effectiveness of our CEMA in filtering out low-informative samples.
>
> We have included these comparisons and discussions in Sections A and E.10 of our revised manuscript.
>
>
> Table A. Comparisons of different schemes to identify unreliable and low-informative samples on ImageNet-C (Gaussian noise, severity level 3).
> | Method| Acc. (%) | # Uploads |
> | --- | --- | --- |
> | SENTRY [B] | 45.6| 17,299|
> | ETA [C] | 50.1| 18,862|
> | CEMA (Ours) | **51.1** | 17,479|
>
> [A] Active Learning for Convolutional Neural Networks: A Core-Set Approach. ICLR 2018.
>
> [B] Sentry: Selective entropy optimization via committee consistency for unsupervised domain adaptation. ICCV 2021.
>
> [C] Efficient Test-Time Model Adaptation without Forgetting. ICML 2022.
>
> ---
>
> >Q3. In equation 6, is $f_{\theta}(x)$ and the pseudo labels $\hat{y}$ the same thing or different?
>
> A3. The model logits $f_{\theta}(x)$ and the pseudo label $\hat{y}$ are **different**. We detail the differences in the following: 1) $f_{\theta}(x)$ is a vector that represents the model logits of the corresponding sample $x$. It is used to calculate the KL loss to align the prediction distribution of the foundation and edge models. 2) $\hat{y}$ is a one-hot vector that denotes the pseudo label of the sample $x$. It is used to calculate the CE loss to align the decision boundaries. Note that $\hat{y}$ can be calculated through $f_{\theta}(x)$ by $\hat{y}=\text{argmax}(f_{\theta}(x))$.
>
> We have made it clearer in Section 2.3 of our revised manuscript.
>
> ---
>
> >Q4. To use KL divergence loss and CE loss together is interesting. I wonder how the hyperparameter alpha and beta change in different dynamic scenarios.
>
> A4. We have examined the impact of the hyperparameters $\alpha$ and $\beta$ on the ImageNet-C (Gaussian noise, severity level 3) dataset (See Tables 10-11 in the supplementary material). The results indicate that the adaptation performance is not sensitive to these hyperparameters. In other scenarios on various datasets, we keep them the same as those on Gaussian noise, including ImageNet-R and ImageNet-C (encompassing 15 types of corruption and 5 severity levels, 75 different scenarios in total). Extensive experiments show that our CEMA works well with these hyperparameters.
>
> We have made it clearer in Section E.2 of our revised manuscript.
>
> ---
>
> We sincerely hope our clarifications above have addressed your questions.

---

> ### Author Response · Authors · 2023-11-21
> **Looking Forward to the Response from Reviewer Q1Z2**
>
> Dear Reviewer Q1Z2,
>
> We express our gratitude for your valuable feedback aimed at enhancing our work. We have provided detailed responses addressing your initial concerns.
>
> We look forward to more discussions with you if you have any outstanding concerns or questions.
>
> Best regards,
>
> The Authors

---

> > ### Author Response · Authors · 2023-11-22
> > **Kind reminder for discussion**
> >
> > Dear Reviewer Q1Z2,
> >
> > We have provided point-by-point responses to your concerns but still haven’t gotten any feedback from you. Do you have any further comments/suggestions?
> >
> > Best regards,
> >
> > The Authors

---

> ### Author Response · Authors · 2023-11-23
> **Follow-up on Author-Reviewer Discussion**
>
> Dear Reviewer Q1Z2,
>
> As the author-reviewer discussion period draws to a close, if you have any remaining questions, please don't hesitate to let us know.
>
> Thank you once again for your time in the review process.
>
> Best regards,
>
> The Authors

---

### Official Review · Reviewer_5VG6 · 2023-10-31

**Soundness:** 3 good
**Presentation:** 2 fair
**Contribution:** 3 good
**Rating:** 8
**Confidence:** 4

**Summary:**

The authors present a Cloud-Edge Model Adaptation (CEMA) paradigm that executes dynamic model adaptation, which puts all adaptation workloads to the cloud and only requires vanilla inference in edges. A replay-based entropy distillation method is also proposed to improve the adaptation performance of the edge model. Extensive experiments show that CEMA achieve SOTA performance with lower communication cost.

**Strengths:**

1. The proposed cloud-edge model adaptation (CEMA) framework seems novel to me.

2. The CEMA paradigm only requires the edges to perform forward computation, which is important considering that backpropagation on edge is difficult.

3. The proposed dynamic unreliable and low-informative sample exclusion are simple but effective.

4. Extensive experiments and ablation studies are provided. There are large performance improvements over previous methods.

**Weaknesses:**

1. The proposed method is only evaluated on classification tasks. Could the proposed method be extended to other tasks such as object detection?

**Questions:**

1. From table 15, it seems that the performance improves as the replay buffer increase. Why not use all the uploaded samples for adaptation?

2. What's the performance if no teacher model is used?

---

> ### Author Response · Authors · 2023-11-20
> **Responses to Reviewer 5VG6**
>
> We deeply appreciate your valuable feedback and constructive comments on improving our work. We would like to address your questions below.
>
> ---
>
> >Q1. The proposed method is only evaluated on classification tasks. Could the proposed method be extended to other tasks such as object detection?
>
> A1. We conduct more experiments by applying our CEMA on corrupted COCO 2017 [A] with the YOLOv5 [B] model. We generate this corrupted version (Gaussian noise, severity level 3) of the COCO 2017 dataset following ImageNet-C. The number of total validation images in corrupted COCO is 5,000. We use YOLOv5-nano as the edge model and YOLOv5-large as the foundation model. From Table A, our CEMA outperforms the baseline (18.3 vs. 11.6 mAP) and ETA (18.3 vs. 15.0 mAP). Moreover, this improved performance was achieved with a smaller number of samples uploaded for adaptation. The results demonstrate the applicability and effectiveness of our CEMA on the object detection task.
>
> We have included these comparisons and discussions in Section E.11 of our revised manuscript.
>
> Table A. Comparisons on corrupted COCO 2017 (Gaussian noise, severity level 3).
>
> | Model                  | mAP     | # Uploads     |
> | ---------------------- | :-------: | :-------------: |
> | YOLOv5-nano (baseline) | 11.6    | --            |
> | $\bullet~$ ETA         | 15.0    | 3,875 (78%)     |
> | $\bullet~$ CEMA (Ours) | **18.3** | **3,261 (65%)** |
>
> [A] Microsoft COCO: Common Objects in Context. arXiv 2014.
> [B] You Only Look Once: Unified, Real-Time Object Detection. CVPR 2016.
>
> ---
>
> >Q2. From table 15, it seems that the performance improves as the replay buffer increase. Why not use all the uploaded samples for adaptation?
>
> A2. We do not use all the uploaded samples for adaptation based on that adaptation accuracy reaches a plateau when the buffer size exceeds 10,000. In Table A, employing a replay buffer of unlimited size does not yield any improvement in adaptation accuracy, which remains at 51.1%. However, this approach leads to a significant increase in storage usage, escalating from 5.6 GB to 9.8 GB. Consequently, we have chosen a buffer size of 10,000 for our experiments.
>
> We have included these comparisons and discussions in Section E.7 in our revised manuscript.
>
> Table A. Effect of the size of the replay buffer on ImageNet-C (Gaussian noise, severity level 3).
>
> | Size         | 0    | 1,000 | 2,000 | 3,000 | 5,000 | 10,000 | $\infty$ |
> | ------------ | ---- | ---- | ---- | ---- | ---- | ----- | -------- |
> | Accuracy (%) | 47.7 | 50.0 | 50.3 | 50.5 | 50.9 | 51.1  | 51.1     |
> | Storage (GB)  | 0    | 0.6  | 1.1  | 1.7  | 2.8  | 5.6   | 9.8      |
>
> ---
>
> >Q3. What's the performance if no teacher model is used?
>
> A3. To verify the effectiveness of the teacher model in distillation, we have conducted experiments on ImageNet-C (Gaussian noise, severity levels 3 and 5). The results have been presented in Table 12 of the supplementary material. From Table A, our CEMA with the teacher model outperforms CEMA without that (51.1% vs. 50.0% in severity level 3). The improvement can be attributed to the teacher model's ability to leverage its knowledge of out-of-distribution data and transfer this knowledge to the edge model. These results collectively demonstrate the effectiveness of the teacher model in enhancing the adaptation accuracy of the edge model.
>
> We have made these comparisons more clear in Section E.3 of the revised manuscript.
>
> Table A. Effect of the teacher model in our CEMA on ImageNet-C (Gaussian noise, severity level 3). Note that the results have already been presented in Table 12 of the supplementary material.
>
> | Teacher Model | Acc. (%) | # Uploads |
> | ------------- | -------- | --------- |
> | $\times$      | 50.0     | 17,263    |
> | $\checkmark$  | 51.5     | 17,479    |
>
> ---
>
> We sincerely hope our clarifications above have addressed your questions.

---

> > ### Comment · Reviewer_5VG6 · 2023-11-22
> > **Comments to author response**
> >
> > Thanks for the detailed clarifications. My concerns are well addressed.

---

> > > ### Author Response · Authors · 2023-11-22
> > > **Thank you for appreciating our Responses!**
> > >
> > > Dear Reviewer 5VG6,
> > >
> > > We are glad to know that our response has addressed your questions. We would like to thank you again for appreciating our work and recognizing our contributions!
> > >
> > > Best,
> > >
> > > The Authors

---

### Official Review · Reviewer_iQ7Q · 2023-11-02

**Soundness:** 3 good
**Presentation:** 4 excellent
**Contribution:** 3 good
**Rating:** 6
**Confidence:** 4

**Summary:**

The paper introduces a Cloud-Edge Model Adaptation (CEMA) paradigm for dynamic model adaptation. This approach delegates adaptation workloads to the cloud, thereby reducing the burden on edge devices. To minimize communication overhead, CEMA excludes unreliable high-entropy and low-informative low-entropy samples from uploading to the cloud. The model leverages knowledge distillation from the foundation model to guide the edge model, and a replay buffer is employed to enhance data utilization efficiency. Experimental results demonstrate a 60% reduction in communication costs compared to state-of-the-art methods on ImageNet-C.

**Strengths:**

* This paper introduces the Cloud-Edge Model Adaptation (CEMA) paradigm, which addresses the dynamic model adaptation problem in a novel way.

* The paper is of high quality, the language is clear, the structure is clean, the related work review in the appendix is adequate (including a valuable comparative analysis with various methods), and the figures are clear and easy to follow.

* The manuscript is very clear in the explanations and the methodology.

* As a practical method for model adaptation, I think this paper is of great chance to benefit the community, especially in real-world scenarios.

**Weaknesses:**

The overall framework appears to be somewhat straightforward as it contains multiple steps. The selection scheme used in the paper is relatively simple, as mentioned in Q1.  Additionally, the selection scheme is designed to exclude data that is either entirely out-of-distribution or absolutely in-distribution. There could be alternative methods to identify these two types of data beyond logits.

**Questions:**

Q1. The authors employ the entropy of the logits to assess uncertainty and selectively upload test samples, excluding both unreliable and low-informative ones. However, recent research has pointed out that neural networks can exhibit overconfidence. In such cases, can the uncertainty of a sample still be accurately evaluated based on the logits?

Q2. The sample selection process involves dynamically adjusting the threshold and incorporating more samples into training. This idea seems to be similar to self-paced learning. Can the authors elaborate on the relationship between self-paced learning and their sample-selection scheme?

---

> ### Author Response · Authors · 2023-11-20
> **Responses to Reviewer iQ7Q [1/2]**
>
> We deeply appreciate your constructive comments. We would like to address your questions below.
>
> ---
>
> >Q1. The overall framework appears to be somewhat straightforward as it contains multiple steps. The selection scheme used in the paper is relatively simple, as mentioned in Q1. Additionally, the selection scheme is designed to exclude data that is either entirely out-of-distribution or absolutely in-distribution. There could be alternative methods to identify these two types of data beyond logits.
>
> A1. Thanks for your valuable feedback. To further demonstrate the effectiveness of our sample identification strategy, we add more experiments on ImageNet-C to compare our CEMA with more existing strategies, namely SENTRY [A] and BALD [B]. SENTRY measures the sample information via the prediction consistency regarding different data augmentations. In addition, BALD achieves this by calculating the entropy differences between the current sample and previous samples.
>
> As presented in Table A, our CEMA outperforms SENTRY, demonstrating a significantly higher accuracy (51.1% vs. 45.6%). This suggests that SENTRY's reliance on prediction consistency is less effective in identifying beneficial samples for entropy minimization. Furthermore, BALD, with an adaptation accuracy of 50.7%, also falls short of the performance achieved by our CEMA. These comparisons underscore CEMA's superior capability in discerning low-informative samples.
>
> We have included these comparisons and discussions in Section E.10 of our revised manuscript.
>
>
> Table A. Comparisons of different strategies to identify unreliable and low-informative samples on ImageNet-C (Gaussian noise, severity level 3).
> | Method      | Acc. (%) | # Uploads |
> | ----------- | -------- | --------- |
> | SENTRY [A]  | 45.6     | 17,299    |
> | BALD [B]    | 50.7     | 17,319    |
> | CEMA (Ours) | **51.1** | 17,479    |
>
>
> [A] Sentry: Selective entropy optimization via committee consistency for unsupervised domain adaptation. ICCV 2021.
>
> [B] Bayesian active learning for classification and preference learning. arXiv 2011.
>
>
> ---
>
> >Q2. The authors employ the entropy of the logits to assess uncertainty and selectively upload test samples, excluding both unreliable and low-informative ones. However, recent research has pointed out that neural networks can exhibit overconfidence. In such cases, can the uncertainty of a sample still be accurately evaluated based on the logits?
>
> A2. Yes, our entropy-based criteria remain effective in identifying unreliable and low-informative samples even with overconfident models. To evaluate the overconfidence in our edge model, we employ a commonly used metric Expected Calibration Error (ECE) [A]. ECE measures the average differences between the model's predicted confidence and its actual accuracy across various confidence intervals. A lower ECE value indicates reduced overconfidence in the model's predictions. Compared with the best counterpart ETA (ECE=6.67%), our CEMA achieves a lower ECE value (3.07%). This substantial reduction in ECE suggests that the edge model in **CEMA exhibits significantly less overconfidence**. Consequently, the uncertainty estimation using entropy in CEMA could potentially be more accurate than that in ETA.
>
> Moreover, it is important to acknowledge that a certain degree of overconfidence is an inherent aspect of neural network models. Despite this, our CEMA demonstrates a robust capability to effectively filter out unreliable and low-informative samples (see results in Tables 1-4). This effectiveness indicates that the impact of overconfidence on our CEMA is limited. Thus, even in the presence of inherent overconfidence in neural networks, the approach adopted by CEMA to assess uncertainty and selectively upload test samples is validated.
>
> We have included these discussions in Section B.5 of our revised manuscript.
>
>
> [A]. Obtaining Well Calibrated Probabilities Using Bayesian Binning. AAAI, 2015.

---

> ### Author Response · Authors · 2023-11-20
> **Responses to Reviewer iQ7Q [2/2]**
>
> Q3. The sample selection process involves dynamically adjusting the threshold and incorporating more samples into training. This idea seems to be similar to self-paced learning. Can the authors elaborate on the relationship between self-paced learning and their sample-selection scheme?
>
> A3. Thank you for your suggestion. We clarify our differences compared with existing self-paced learning methods as follows.
>
> Self-paced learning [A,B,C,D] automatically reorders samples during training based on their difficulty. For instance, SPLD[B] pre-cluster the training data and encourages balance samples section from different clusters. Despite both self-paced learning and CEMA improving learning efficiency by active sample selection, self-paced learning focuses on learning robustness instead of mitigating computation cost. Consequently, it differs from CEMA in two aspects:
>
>  - Self-paced learning initiates the learning process with an easy subset of samples, which includes even those of low informativeness. In contrast, our CEMA approach specifically focuses on selecting samples that are both informative and reliable for adaptation. These two kinds of samples are beneficial to the adaptation process.
>  - Self-paced learning offline selects the whole dataset for training as the remaining samples become easier. While our CEMA dynamically adjusts the threshold to continually filter out less reliable samples. This process is conducted online, with a keen focus on maintaining communication efficiency.
>
> We have included these discussions in the related work in Section A of our revised manuscript.
>
> [A] Self-paced learning for latent variable models. NeurIPS, 2010.
>
> [B] Self-Paced Learning with Diversity. NeurIPS, 2014.
>
> [C] Self paced deep learning for weakly supervised object detection. TPAMI, 2018.
>
> [D] Self-paced contrastive learning for semi-supervised medical image segmentation with meta-labels. NeurIPS, 2021.
>
> ---
>
> We sincerely hope our clarifications above have addressed your questions.

---

### Official Review · Reviewer_xxok · 2023-11-03

**Soundness:** 4 excellent
**Presentation:** 3 good
**Contribution:** 4 excellent
**Rating:** 6
**Confidence:** 4

**Summary:**

This paper introduces a novel learning paradigm aimed at enhancing the adaptability of cloud-edge models to address challenges posed by out-of-distribution test samples in real-world scenarios. The proposed approach is both practical and holds substantial significance. Specifically, to reduce communication overhead, the authors have incorporated a dynamic sample filtering strategy, allowing for the identification and exclusion of unreliable and low-informative samples. Furthermore, to further augment the edge model's capabilities and fully capitalize on the abundant cloud resources, the authors have integrated a substantial foundational model to serve as a guiding teacher for the edge model. Extensive experimental results on ImageNet-C and ImageNet-R datasets serves to underscore the efficacy of the presented method.

**Strengths:**

1.	The proposed collaborative cloud-edge model adaptation (CEMA) paradigm addresses a highly practical problem in the realm of cloud-edge model deployment, emphasizing the challenges of distribution shifts and the limited resources of the edge devices.
2.	The proposed CEMA paradigm is a pioneering achievement in the field. It effectively divides adaptation tasks and distributes them between the cloud and edge devices, resulting in optimized resource utilization and the assurance of robust performance.
3.	The experimental results demonstrate that the proposed method not only achieves the highest level of out-of-distribution performance but also reduces communication costs by an impressive 60% when compared to SOTAs on the ImageNet-C and ImageNet-R benchmarks.
4.	The paper is well-written and easy to follow. Furthermore, it is accompanied by illustrative figures that enhance its overall readability.

**Weaknesses:**

1.	In Section Identification on low-informative samples, the author claims that ‘We emphasize that uploading samples does not block the edge from inferring on next incoming samples. In other words, the processes of inference and uploading can be executed simultaneously.’. How do the authors decide which test samples use which updated model to make a prediction? More explanations are required.
2.	In Equation (6), the foundation model assigns pseudo labels to the uploaded samples. Simultaneously, the authors employ entropy to update the model, introducing another pseudo label (the maximum value). Is there a potential conflict between these approaches? What consequences might arise if the entropy loss is eliminated?
3.	In Algorithm 1, line 3, ‘Calculate S(X) via Eqn. (x)’, It is confused what is Eqn.(x).

**Questions:**

See weakness.

---

> ### Author Response · Authors · 2023-11-20
> **Responses to Reviewer xxok**
>
> We are grateful for your time and effort. We would like to answer your questions below.
>
> ---
>
> >Q1. In Section Identification on low-informative samples, the author claims that ‘We emphasize that uploading samples does not block the edge from inferring on next incoming samples. In other words, the processes of inference and uploading can be executed simultaneously.’. How do the authors decide which test samples use which updated model to make a prediction? More explanations are required.
>
> A1. In the edge device, **we would infer a sample for only one time via the current edge model**. Specifically, we feed a sample into the edge model and then determine whether this sample would be uploaded to the cloud based on Eqn. (4). In the cloud, once receiving a batch of $N$ uploaded samples, the edge model would be adapted for one time via Eqn. (6). After adaptation, the edge model would update the parameters from the cloud. Then the subsequent samples are processed using this newly updated edge model.
>
> We have also taken into account scenarios of poor network connectivity (see Table 18 of the supplementary material). In this case, the edge model would be adapted for $K$ time ($K$>1) via Eqn. (6) in the cloud. The parameters updated through these adaptations are then sent back to the edge, ensuring that the next incoming sample is inferred using the most recently updated edge model.
>
> We have included these discussions in Section B.4 of our revised manuscript.
>
>
> ---
>
> >Q2. In Equation (6), the foundation model assigns pseudo labels to the uploaded samples. Simultaneously, the authors employ entropy to update the model, introducing another pseudo label (the maximum value). Is there a potential conflict between these approaches? What consequences might arise if the entropy loss is eliminated?
>
>
> A2. **There are no conflicts between the entropy minimization and cross entropy (via pseudo labels)**, since they contribute different effects in our CEMA:
>
> - **Cross Entropy with Pseudo Labels**: We use the cross entropy via pseudo labels to provide direct label guidance for a single sample to align the decision boundary.
> - **Entropy Minimization**: We optimize the average entropy (also namely uncertainty) over a batch of samples to improve the overall model confidence regarding its predictions, rather than only encouraging a single sample to converge towards its highest predicted probability.
>
> **Empirical Comparisons**. In Table A, we conduct ablation studies to investigate the effectiveness of the entropy loss and the CE loss. Both the cross entropy and entropy losses work together to boost the overall cloud-edge model adaptation performance. From the results, our CEMA with CE Loss achieves better adaptation accuracy than CEMA without that. Besides, while we employ both CE loss and Entropy loss, the adaptation performance further increases. The experimental results demonstrate these losses are complementary and effective in our CEMA.
>
> We have included these comparisons and discussions in Section E.3 in our revised manuscript.
>
>
> Table A. Effect of components in the loss for updating the edge model on ImageNet-C (Gaussian noise, severity level 3).
> | CE Loss      | Ent Loss     | Acc. (%) | # Uploads |
> | ------------ | ------------ | -------- | --------- |
> | $\times$     | $\checkmark$ | 50.5     | 17,379    |
> | $\checkmark$ | $\times$     | 50.2     | 18,497    |
> | $\checkmark$ | $\checkmark$ | 51.5     | 17,479    |
>
> ---
>
> >Q3. In Algorithm 1, line 3, ‘Calculate $S(X)$ via Eqn. (x)’, It is confused what is Eqn.(x).
>
> A3.Thank you for your valuable suggestions. We have addressed the typo in the revised manuscript, and it should now correctly reference "Eqn. (4)."
>
> ---
>
> We sincerely hope our clarifications above have addressed your concerns.

---

### Author Response · Authors · 2023-11-21
**General Responses**

Dear ACs and Reviewers,

We extend our sincere gratitude for your valuable time and insightful feedback on our paper. Your comments have been instrumental in refining our work. In addition to our specific responses to each reviewer, we would like to 1) express our gratitude for your recognition of our work, and 2) emphasize the major modifications made in our revised manuscript.

1. **We are encouraged by your acknowledgment of the novelty and contributions of our work**.
   - “The proposed CEMA paradigm addresses a highly **practical dynamic model adaptation problem** in a **novel** way”. [Reviewers xxok, iQ7Q, 5VG6]
   - “The proposed CEMA paradigm is a **pioneering** achievement in the field, which is of **great chance to benefit the community**, especially in real-world scenarios”. [Reviewers xxok, iQ7Q]
   - “The proposed dynamic unreliable and low-informative sample exclusion are **simple but effective**, resulting in optimized resource utilization and the assurance of robust performance”. [Reviewers xxok, iQ7Q, 5VG6]
   - “The experiments are **comprehensive**, demonstrating **large performance improvements** over SOTAs”. [Reviewers xxok, iQ7Q, 5VG6, Q1Z2]
   - “The paper is **well-written and easy to follow** in the explanations and the methodology”. [Reviewers xxok, iQ7Q, Q1Z2]

2. **We summarize the main modifications in our revised paper (highlighted in blue)**.
   - We add discussions regarding the parameter updating mechanisms (see Appendix B.4 in the revised manuscript) and complementarity of the entropy minimization and the cross entropy loss (see Appendix E.3 in the revised manuscript). [Reviewer xxok]
   - We add discussions regarding the applicability of our proposed entropy-based criteria with the overconfident model in Appendix B.5. [Reviewer iQ7Q]
   - We add discussions regarding the hyperparameters $\alpha$ and $\beta$ Appendix E.2, showing that these hyperparameters are able to be applied to various scenarios on different datasets. [Reviewer Q1Z2]
   - We add more discussions regarding the differences between our CEMA and self-paced learning as well as low-informative sample identification in Appendix A. [Reviewers 5VG6, Q1Z2]
   - We add more results on the object detection task in Appendix E.11. The new results further verify the effectiveness of our proposed CEMA. We add more ablation results on our CEMA in the context of an unlimited replay buffer and absence of a foundation model as the teacher in Appendix E.7 and E.3, respectively.  [Reviewer 5VG6]


Best regards,

The Authors

---

### Meta-Review · Area_Chair_U3S1 · 2023-12-05

**Metareview:**

The Cloud-Edge Model Adaptation (CEMA) paradigm introduced in this paper offers a novel approach for dynamically adapting deep learning models on resource-constrained edge devices. CEMA shifts the adaptation workload to the cloud, reducing edge devices' computational load. It employs a selective sample filtering strategy to minimize communication overhead by excluding unreliable and low-informative data. Additionally, the paradigm incorporates a replay-based entropy distillation method, utilizing a foundational cloud model as a guiding teacher for the edge model.

**Justification For Why Not Higher Score:**

The reviewers raised concerns about the discussion and comparison with existing approaches for identifying low-informative samples. The authors have directly addressed these concerns, clarifying that while identifying low-informative samples is an important aspect of their work, it is only one component of the broader Cloud-Edge Model Adaptation (CEMA) paradigm they propose. To respond to the reviewers' inquiries, the authors included a brief comparison with existing strategies in active learning and domain adaptation.

**Justification For Why Not Lower Score:**

This paper addresses the critical challenges of distribution shifts and resource limitations in cloud-edge model deployment. The CEMA paradigm stands out for its effective distribution of adaptation tasks between cloud and edge devices, optimizing resource use while ensuring robust performance.

---

### Decision · Program_Chairs · 2024-01-16

Accept (poster)